# Precise optical control of gene expression in *C elegans* using improved genetic code expansion and Cre recombinase

Lloyd Davis[1], Inja Radman[2†], Angeliki Goutou[1†], Ailish Tynan[1], Kieran Baxter[1], Zhiyan Xi[1], Jack M O'Shea[1], Jason W Chin[2*], Sebastian Greiss[1*]

[1]Centre for Discovery Brain Sciences, University of Edinburgh, Edinburgh, United Kingdom; [2]Medical Research Council Laboratory of Molecular Biology, Cambridge, United Kingdom

**Abstract** Synthetic strategies for optically controlling gene expression may enable the precise spatiotemporal control of genes in any combination of cells that cannot be targeted with specific promoters. We develop an improved genetic code expansion system in *Caenorhabditis elegans* and use it to create a photoactivatable Cre recombinase. We laser-activate Cre in single neurons within a bilaterally symmetric pair to selectively switch on expression of a loxP-controlled optogenetic channel in the targeted neuron. We use the system to dissect, in freely moving animals, the individual contributions of the mechanosensory neurons PLML/PLMR to the *C. elegans* touch response circuit, revealing distinct and synergistic roles for these neurons. We thus demonstrate how genetic code expansion and optical targeting can be combined to break the symmetry of neuron pairs and dissect behavioural outputs of individual neurons that cannot be genetically targeted.

**\*For correspondence:**
chin@mrc-lmb.cam.ac.uk (JWC);
s.greiss@ed.ac.uk (SG)

†These authors contributed equally to this work

**Competing interest:** The authors declare that no competing interests exist.

## Introduction

Genetic code expansion offers the ability to introduce, during translation, functionalities into proteins beyond those offered by the set of canonical, naturally occurring amino acids. This is achieved by providing orthogonal translation components: an orthogonal aminoacyl-tRNA synthetase (aaRS) paired with a cognate orthogonal tRNA, and assignment of a codon to determine the site of incorporation of a non-canonical amino acid (ncAA) (*Chin, 2017*). The genetic code of several single and multicellular systems has been expanded to date (*Brown et al., 2018a*). *Caenorhabditis elegans* was the first multicellular organism in which genetic code expansion was established, and the incorporation of ncAA has been demonstrated in multiple *C. elegans* tissues (*Greiss and Chin, 2011*; *Parrish et al., 2012*). Currently available orthogonal systems, however, often lack the efficiency required to fully harness the possibilities offered by genetic code expansion, especially in multicellular systems. A large variety of non-canonical amino acids are available for in vivo use, among them photocaged amino acids, which can be used to confer light-activated control over the function of proteins in a host organism (*Courtney and Deiters, 2018*).

Sequence-specific DNA recombinases, such as Cre and FLP, have become transformative for controlling gene expression in many systems, including worms, flies, and vertebrates, by irreversibly activating or deactivating target gene expression. Indeed, the photo-induced reconstitution of split recombinases (*Schindler et al., 2015*; *Kawano et al., 2016*) might be used to optically target recombinase activity to a single cell. However, the sensitivity of this technique can be hampered by high background levels of light-independent recombinase activity. Moreover, these systems are induced using blue light, rendering their use incompatible with most optogenetic and imaging applications, and requiring cells to be kept in the dark, complicating their use in transparent model

**eLife digest** Animal behaviour and movement emerges from the stimulation of nerve cells that are connected together like a circuit. Researchers use various tools to investigate these neural networks in model organisms such as roundworms, fruit flies and zebrafish. The trick is to activate some nerve cells, but not others, so as to isolate their specific role within the neural circuit.

One way to do this is to switch genes on or off in individual cells as a way to control their neuronal activity. This can be achieved by building a photocaged version of the enzyme Cre recombinase which is designed to target specific genes. The modified Cre recombinase contains an amino acid (the building blocks of proteins) that inactivates the enzyme. When the cell is illuminated with UV light, a part of the amino acid gets removed allowing Cre recombinase to turn on its target gene.

However, cells do not naturally produce these photocaged amino acids. To overcome this, researchers can use a technology called genetic code expansion which provides cells with the tools they need to build proteins containing these synthetic amino acids. Although this technique has been used in live animals, its application has been limited due to the small amount of proteins it produces. Davis et al. therefore set out to improve the efficiency of genetic code expansion so that it can be used to study single nerve cells in freely moving roundworms.

In the new system, named LaserTAC, individual cells are targeted with UV light that 'uncages' the Cre recombinase enzyme so it can switch on a gene for a protein that controls neuronal activity. Davis et al. used this approach to stimulate a pair of neurons sensitive to touch to see how this impacted the roundworm's behaviour. This revealed that individual neurons within this pair contribute to the touch response in different ways. However, input from both neurons is required to produce a robust reaction.

These findings show that the LaserTAC system can be used to manipulate gene activity in single cells, such as neurons, using light. It allows researchers to precisely control in which cells and when a given gene is switched on or off. Also, with the improved efficiency of the genetic code expansion, this technology could be used to modify proteins other than Cre recombinase and be applied to other artificial amino acids that have been developed in recent years.

systems. Recently, genetic code expansion methods were used to create photocaged versions of Cre in cultured mammalian cells and zebrafish embryos (*Luo et al., 2016*; *Brown et al., 2018b*). In photocaged Cre, an orthogonal aaRS/tRNA$_{CUA}$ pair is used to replace a catalytically critical tyrosine or lysine residue in the active site of Cre with a photocaged tyrosine or lysine residue, resulting in a catalytically inactive enzyme (*Luo et al., 2016*; *Brown et al., 2018b*). The photocaging group can then be removed through a short exposure to 365 nm UV light, activating Cre recombinase, which in turn activates expression of its target genes. We set out to create an optimised photocaged Cre system that enables the optical activation of gene expression in single cells in an animal. The precise spatiotemporal control of gene expression offered by photocaged Cre would have a wide range of applications by allowing the expression of genes of interest in single cells or combinations of cells. Applied to control, for example, optogenetic channels or other neuromodulatory proteins, it could be a transformative tool in neurobiology.

Defining the molecular and cellular basis of behaviour is a fundamental challenge in neuroscience. *C. elegans* contains 302 neurons, for which the physical connectivity has been determined (*White et al., 1986*). This physical map provides a conceptual framework within which to design experiments to understand the signal processing properties of neural circuits that define behaviour. Central to this endeavour are methods for the spatiotemporal control of gene expression that enable precise perturbations to be effected in individual cells or combinations of cells and thereby advance our understanding of cellular contributions to circuit function. Two thirds of the neurons in *C. elegans* exist in bilaterally symmetric (left and right) pairs, and similar gross morphological symmetries are seen in the nervous systems of many other animals. A central question is whether and how this morphological symmetry is broken to produce functional asymmetry (*Troemel et al., 1999*; *Wes and Bargmann, 2001*; *Suzuki et al., 2008*; *Hobert et al., 2002*).

Tissue-specific promoters (alone or in combination) can be used to target groups of cells. However, this approach offers only limited precision, and in many cases appropriate promoters are unavailable;

especially in scenarios when investigators aim to limit expression to a single cell or a defined subset of cells. This is particularly limiting in neuroscience studies where the goal is to couple neuronal activity of single neurons or a defined subset of neurons with functional output. Where specific promoters for targeting individual neurons are unavailable, laser ablation can be used to remove specific neurons, but this technology is not compatible with studying intact circuits. Optogenetic approaches using targeted illumination have been applied to stimulate groups or pairs of cells, and track resulting behavioural outputs in freely moving *C. elegans* (*Leifer et al., 2011*; *Stirman et al., 2011*; *Kocabas et al., 2012*). However, without a means of precisely defining expression of optogenetic tools, this approach is technically very challenging as it requires specialised software and hardware tools for real-time tracking and targeting and does not lend itself to isolating single cells in neuronal pairs. Similar obstacles are encountered in other systems, where the solution often also requires technically challenging approaches utilising targeted illumination (*Chen et al., 2018*).

Here we describe the optimisation of genetic code expansion for ncAA incorporation and the optimisation of a photocaged Cre recombinase variant (optPC-Cre) in *C. elegans*. The combination

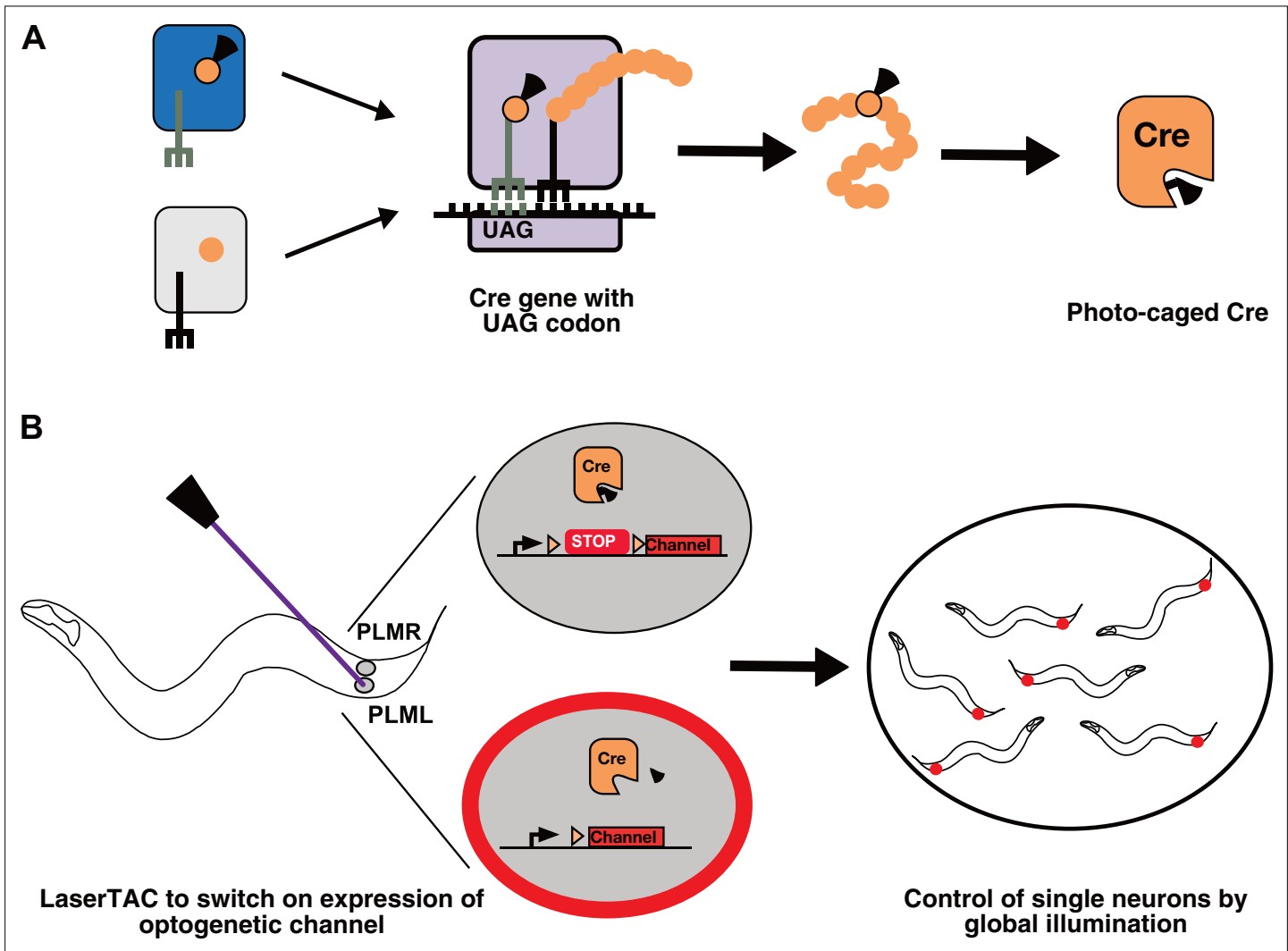

**Figure 1.** Genetic code expansion and LaserTAC. (**A**) A non-canonical amino acid, e.g. photocaged lysine (PCK), is charged onto the orthogonal tRNA$_{CUA}$ by the orthogonal aminoacyl-tRNAsynthetase (top left). The orthogonal components do not interact with the native cellular amino acids, tRNAs, or aminoacyl-tRNA-synthetases (bottom left). After charging, PCK is incorporated into the expanding polypeptide chain in response to an amber stop codon (UAG) during ribosomal translation. The resultant polypeptide is released following translation resulting in full-length photocaged Cre containing the PCK residue. (**B**) Photocaged Cre recombinase is activated cell-specifically in single *C. elegans* neurons by uncaging the PCK in its active site using a 365 nm laser. Activated Cre switches on expression of an optogenetic channel. Global illumination of freely moving animals can then be used to control cells expressing the optogenetic channel.

of these advances allows us to optically control gene expression with unprecedented precision and over 95 % efficiency in a population of animals (compared to less than 1 % efficiency before optimisation). By using a microscope-mounted 365 nm laser, we photo-activate Cre and thereby control gene expression in single cells (*Figure 1*). We name this approach **L**aser **T**argeted **A**ctivation of **C**re-Lox recombination (LaserTAC). LaserTAC is fast and within the technical capabilities of any lab currently able to perform laser ablation, allowing easy generation, in a single session, of dozens of animals with defined expression patterns for use in downstream experiments. We demonstrate the utility of LaserTAC by using it to target the expression of optogenetic channels to individual *C. elegans* touch sensory neurons within a left/right pair (PLML and PLMR). These individual neurons cannot be targeted by other methods, and our approach allows us to study their contribution to behaviour for the first time. Our results reveal that the PLM neurons act in synergy to produce a robust touch response requiring input from both neurons. Furthermore, the individual neurons within this pair make asymmetric contributions to the touch response, suggesting distinct roles for PLMR and PLML in the habituation to repeated stimulation.

## Results

### Improved ncAA incorporation

Efficient genetic code expansion depends on the ability of the orthogonal aminoacyl-tRNA synthetase to aminoacylate its cognate tRNA$_{CUA}$, which in turn delivers the ncAA to the ribosome for incorporation in response to the UAG amber stop codon (*Figure 1A*). The charging of the ncAA onto the tRNA$_{CUA}$ occurs mainly in the cytoplasm, and efficient charging is therefore dependent on cytoplasmic aminoacyl-tRNA synthetase availability.

The orthogonal pyrrolysyl-tRNA synthetase (PylRS)/tRNA(Pyl) pair from archeal *Methanosarcina* species is the most versatile and widely used pair for genetic code expansion (*Wan et al., 2014*; *Brown et al., 2018a*). The PylRS/tRNA(Pyl) pair is functional in *C. elegans*, and variants exist that recognise photocaged amino acids (*Greiss and Chin, 2011*; *Gautier et al., 2010*).

When expressed in eukaryotic cells, PylRS is localised predominantly to the nucleus due to a stretch of positively charged amino acids in the N-terminal domain, which is interpreted as a nuclear localisation sequence by the eukaryotic nuclear import machinery. Accordingly, addition of a rationally designed strong nuclear export sequence (S-NES) to the PylRS N-terminus was reported to significantly increase the efficiency of ncAA incorporation in mammalian cells by increasing the amount of cytoplasmic PylRS (*Nikić et al., 2016*).

To test this approach in *C. elegans*, we added the reported S-NES to the N-terminus of PCKRS, a PylRS variant optimised for incorporating photocaged lysine (PCK) (*Gautier et al., 2010*; *Figure 2A*). We used the ubiquitous *C. elegans* promoter *sur-5p* (*Yochem et al., 1998*) to drive S-NES::PCKRS expression and *rpr-1p* (*Parrish et al., 2012*) to drive tRNA(Pyl)$_{CUA}$ expression. To assay PCK incorporation efficiency, we used a dual-colour incorporation sensor; a ubiquitously expressed GFP::mCherry fusion, with the two fluorophore coding sequences separated by an amber stop codon (*Greiss and Chin, 2011*; *Figure 2B*). For this reporter, translation in the absence of PCK leads to termination at the amber stop codon, resulting in production of only GFP. Conversely, in the presence of PCK, tRNA(Pyl)$_{CUA}$ is charged with the ncAA and acts as a nonsense suppressor, resulting in production of the full-length GFP::mCherry fusion protein, which includes a nuclear localisation sequence and an HA tag at its C-terminus. However, we did not detect any significant difference in incorporation between worms expressing unmodified PCKRS and S-NES::PCKRS. In fact, incorporation levels when using the S-NES::PCKRS construct appeared to be lower than with unmodified PCKRS (constructs 'no NES' and 'S-NES'; *Figure 2C*, *Figure 2—figure supplement 1A*).

To test whether S-NES::PCKRS and unmodified PCKRS localise as predicted, we expressed S-NES::PCKRS::GFP and PCKRS::GFP fusion proteins. As expected, the S-NES::PCKRS::GFP fusion localised to the cytoplasm, while unmodified PCKRS::GFP was almost entirely nuclear, reflecting that the S-NES was able to efficiently shift PCKRS localisation from the nucleus to the cytoplasm (constructs 'no NES' and 'S-NES' *Figure 2D*, *Figure 2—figure supplement 1B*). Therefore, we hypothesised that the S-NES tag itself might impinge on PCK incorporation efficiency. We next performed a mini-screen of three nuclear export sequences from human proteins: p120cts-NES, PKIα-NES, and Smad4-NES,

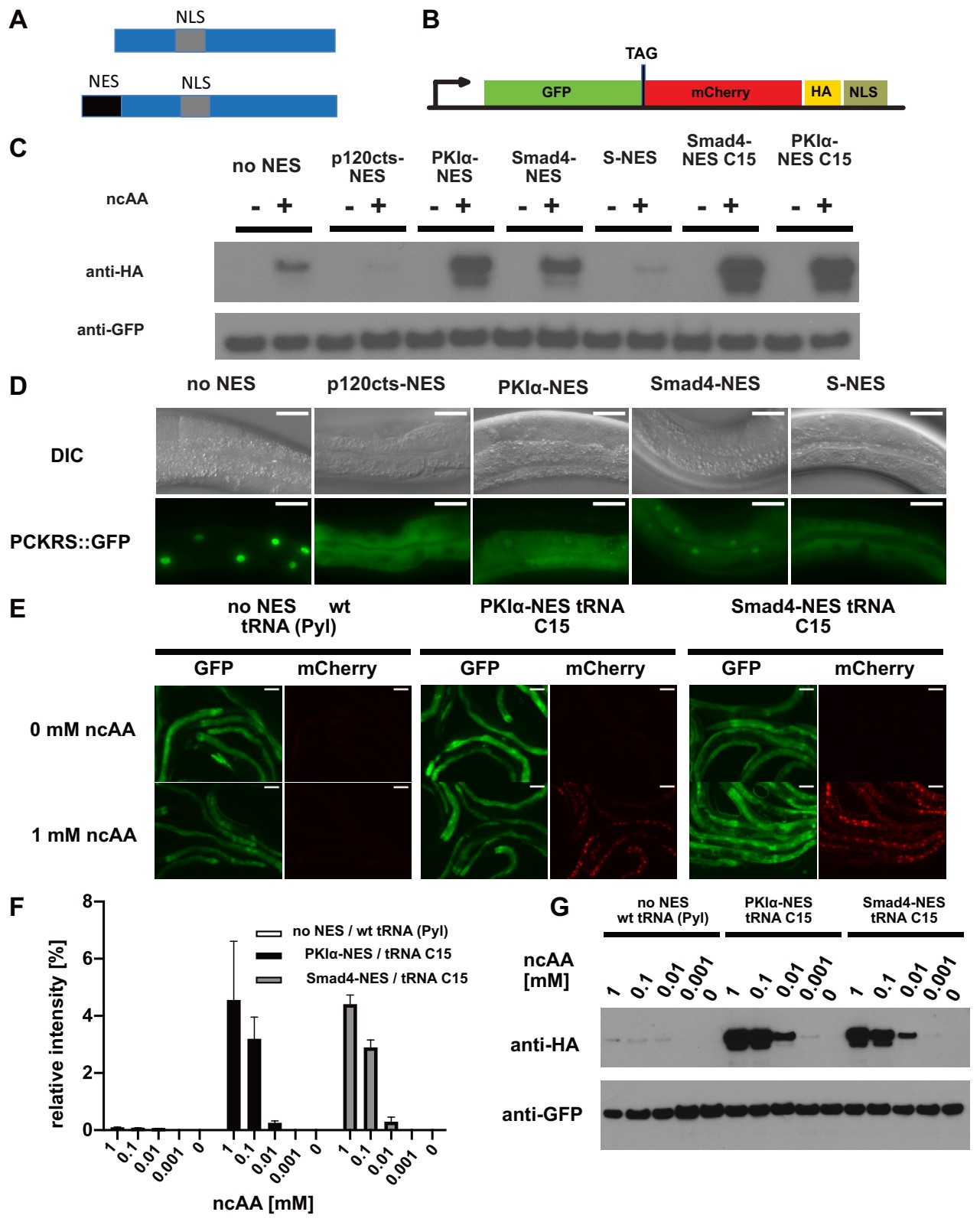

**Figure 2.** Efficiency of genetic code expansion is enhanced in *C. elegans* by use of NES::PCKRS variants and tRNA(C15). (**A**) Wildtype PylRS/PCKRS (top) contains an internal nuclear localisation sequence (NLS) which targets it to the nucleus. A strong nuclear export sequence (NES) can be added to PylRS/PCKRS to shift it to the cytoplasm (bottom). (**B**) The GFP::mCherry reporter has an intersubunit linker containing an amber stop codon (TAG) to direct incorporation of ncAA. The C-terminal NLS moves full-length product to the nucleus, providing a second visual readout, whilst the C-terminal HA

*Figure 2 continued on next page*

*Figure 2 continued*

tag provides a target for western blotting of full-length protein. (**C**) Western blot (anti-HA) for full-length reporter produced using PCKRS with different NES attached to the N-terminus. For comparison, the wildtype PCKRS without NES and the two best NES variants together with optimised tRNA(C15) are also shown. + or – indicates the respective presence or absence of 1 mM ncAA. Samples were normalised using anti-GFP. Quantitative western blots of the lines used are shown in *Figure 2—figure supplement 1A*. (**D**) Nuclear localisation of NES variants fused to PCKRS. Localisation visualised by imaging of a GFP protein directly fused to each NES::PCKRS variant. Scale bars 30 μm. Nuclear to cytoplasmic ratios for the wild-type PCKRS::GFP fusion and two independent lines for each NES variant are shown in *Figure 2—figure supplement 1B*. (**E**) Fluorescent images of randomly selected worms grown in the presence or absence of 1 mM non-canonical amino acid (ncAA). GFP indicates expression of reporter construct, mCherry indicates the presence of full-length reporter protein. Scale bars 80 μm. Enlarged versions of the images for PKIa-NES and Smad4-NES are shown in *Figure 2—figure supplement 1C*. (**F**) Quantification of western blots of wild-type PCKRS ('no NES')/tRNA(Pyl) vs. both PKIα-NES::PCKRS/tRNA(C15) and Smad4-NES::PCKRS/tRNA(C15) at different ncAA concentrations; anti-GFP was used to detect both the full-length GFP;;mCherry protein as well as the GFP truncated at the amber stop. The graphs show the relative intensities of the full-length GFP signal vs. the signal of the GFP truncated at the amber stop codon. Graphs represent the average of two lines and three independent experiments per condition, each experiment was blotted twice. (**G**) Conditions shown in (**F**) probed with anti-HA and anti-GFP. In (**G**), only one line was blotted for each condition.

The online version of this article includes the following source data and figure supplement(s) for figure 2:

**Source data 1.** Band intensities for the quantitative western blots shown in *Figure 2F*.

**Figure supplement 1.** Nuclear export sequences reduce nuclear localisation of PCKRS::GFP and increase ncAA incorporation efficiency.

**Figure supplement 1—source data 1.** Band intensities for the quantitative western blots shown in *Supplementary file 1A*.

which are demonstrated to act as NESs in *C. elegans* (*Yumerefendi et al., 2015*). All tested NESs achieved nuclear export of the synthetase PCKRS (*Figure 2D*, *Figure 2—figure supplement 1B*).

However, the NES variants differentially affected incorporation efficiencies: when we co-expressed NES::PCKRS variants with tRNA(Pyl) and the GFP::mCherry incorporation reporter, we found that the p120cts-NES variant reduced incorporation efficiency below the level of unmodified PCKRS, whereas the PKIα-NES and the Smad4-NES robustly increased PCK incorporation efficiency (*Figure 2C*, *Figure 2—figure supplement 1A*). We selected the two most efficient PCKRS variants for subsequent optimisation experiments.

We then turned to the optimisation of tRNA(Pyl). tRNA(Pyl) contains secondary structure elements not present in canonical mammalian tRNAs. These features reduce compatibility with the endogenous translational machinery, which limits ncAA incorporation in mammalian cells. However, ncAA incorporation efficiency in cultured mammalian cells can be improved by introducing mammalian tRNA elements into the archaeal tRNA(Pyl) or by using engineered mitochondrial tRNAs (*Serfling et al., 2018*). To investigate whether the same approach might improve ncAA incorporation efficiency in other eukaryotes, we tested an improved tRNA variant in the *C. elegans* system. Specifically, we co-expressed the mitochondrial tRNA based variant C15, previously validated in mammalian cells (*Serfling et al., 2018*) together with the GFP::mCherry incorporation reporter, and the two most efficient synthetase variants described above (namely Smad4-NES::PCKRS or PKIα-NES::PCKRS). We found that the presence of the C15 variant significantly increased incorporation efficiency (*Figure 2C, E, F and G*, *Figure 2—figure supplement 1A,C*). Compared to the unmodified PCKRS/tRNA(Pyl) pair, the Smad4-NES::PCKRS/tRNA(C15) and PKIα-NES::PCKRS/tRNA(C15) pairs improved incorporation efficiency by more than 50-fold; from <0.1% to 4.6% and 4.4 %, respectively (*Figure 2F*). We performed all further experiments using the Smad4-NES::PCKRS/tRNA(C15) pair.

## Expression of photocaged Cre recombinase

We next applied our improved incorporation system to express photocaged Cre recombinase (PC-Cre) in the N2 wildtype *C. elegans* laboratory strain. Cre recombinase can be photocaged by replacing K201, a lysine residue in the Cre active site, critical for enzyme activity, with PCK (*Gibb et al., 2010*; *Luo et al., 2016*; *Figure 3A and B*). PC-Cre has previously been expressed in both mammalian cell culture and zebrafish embryos using transient transfection and direct mRNA/tRNA injections (*Luo et al., 2016*; *Brown et al., 2018b*). To direct incorporation of PCK, we generated a PC-Cre construct where we replaced the lysine codon at position 201 in the Cre sequence with an amber stop codon (TAG). The PC-Cre construct was expressed in an artificial operon with GFP, allowing us to visualise its expression. We used as a Cre recombinase target gene a fluorescently tagged channelrhodopsin ChR2::mKate2, separated from its promoter by a transcriptional terminator flanked by loxP sites (*Figure 3C*).

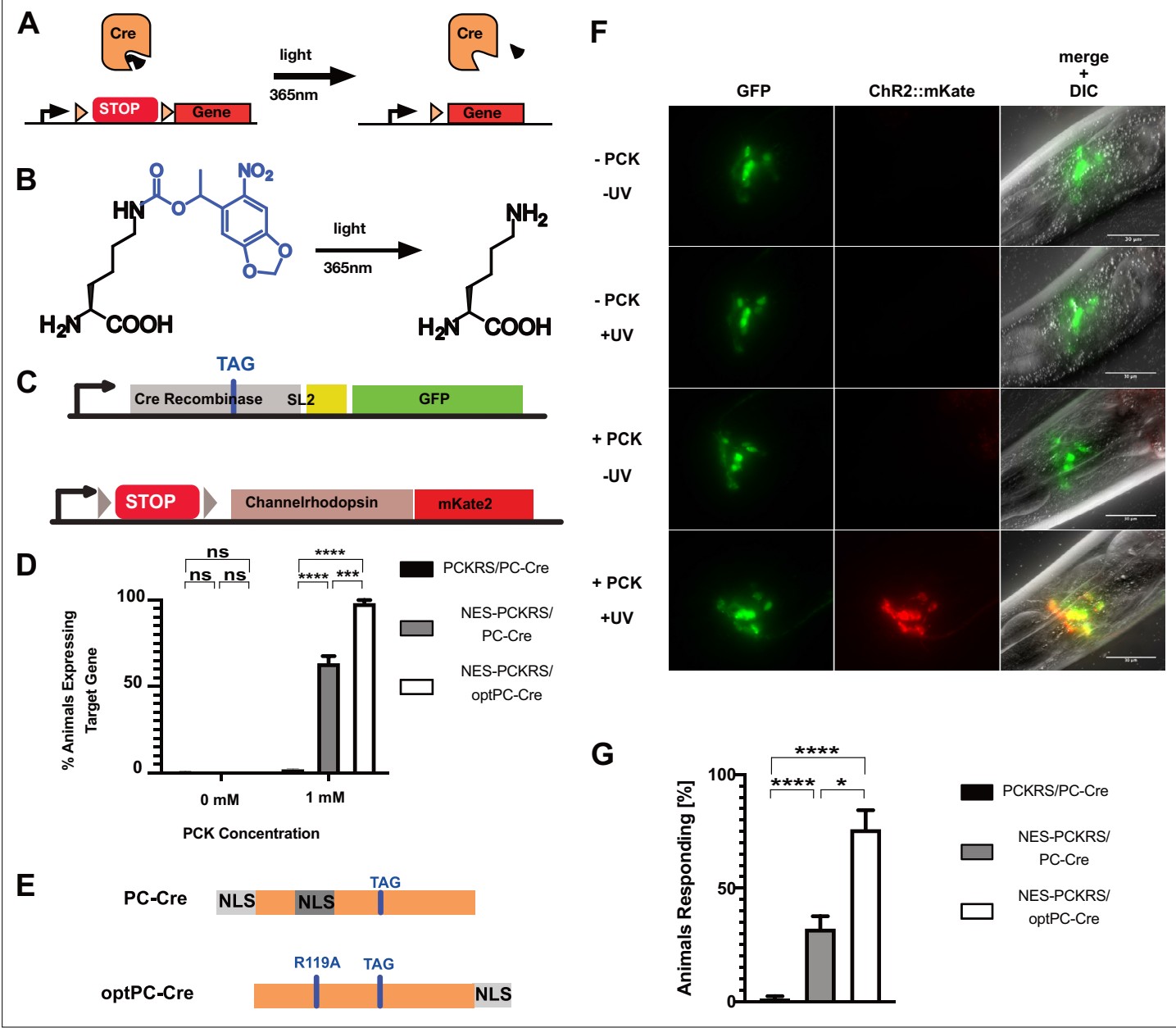

**Figure 3.** Optimisation of PC-Cre recombinase for genetic code expansion in *C. elegans*. (**A**) Cre recombinase can be photocaged by incorporating a photocaged non-canonical amino acid (ncAA) into the active site (black wedge). Transcription of a target gene is blocked by placing a transcription terminator sequence ('STOP') between the gene and its promoter. The terminator sequence is flanked by loxP sites (orange triangles). Upon illumination at 365 nm, the photocaging group is removed and the uncaged, active Cre removes the transcriptional terminator. (**B**) 6-nitropiperonyl-L-Lysine, 'photocaged lysine' (PCK), is a lysine residue with a photocaging group on the side chain. The photocaging group is removed at 365 nm. (**C**) Genetic constructs for PC-Cre controlled expression of channelrhodopsin. See also *Figure 3—figure supplement 1*, . (**D**) Cre activation efficiency in animals with optimised vs. original constructs. Comparison of original PCKRS and tRNA(Pyl)$_{CUA}$ ('PCKRS'), Smad4-NES::PCKRS and tRNA(C15) ('NES-PCKRS'), original photocaged Cre recombinase ('PC-Cre'), and optimised photocaged Cre ('optPC-Cre'). Three experiments were performed using two independent lines for each condition. In each experiment, 30 animals were visually scored for ChR2::mKate2 expression for each condition. Significance determined by Welch's t test. The error is the SEM (n = 6). (**E**) Schematic showing PC-Cre and optPC-Cre. Substitution mutations shown as blue bars. The TAG mutation denotes the site of PCK incorporation. PC-Cre contains both an N-terminal and internal nuclear localisation sequence (NLS). OptPC-Cre contains only a C-terminal NLS, the R119A mutation disables the internal NLS. (**F**) Imaging of worms expressing Smad4-NES::PCKRS, tRNA(C15), and optPC-Cre. Expression of ChR2::mKate2 is dependent on supplementation of PCK and uncaging by exposure to UV light. Scale bars 30 μm. (**G**) Percentage of animals reversing in response to a blue light pulse. 'PCKRS': original PCKRS and tRNA(Pyl)$_{CUA}$; 'NES-PCKRS': Smad4-NES::PCKRS and tRNA(C15); 'PC-Cre': original photocaged Cre recombinase; 'optPC-Cre': optimised photocaged Cre. Assays were carried out using two independent

*Figure 3 continued on next page*

*Figure 3 continued*
lines and >10 animals for each genotype; the mean of three stimulations was determined. The graph shows the mean of two experiments. Significance obtained by Mann–Whitney U test. The error is the SEM (n = 4). *p<0.05, ***p<0.001, ****p<0.0001.

The online version of this article includes the following figure supplement(s) for figure 3:

**Source data 1.** Source data for *Figure 3F,G*.

**Figure supplement 1.** Genetic constructs for expression of photoactivatable Cre and assays to determine Cre activation.

We generated transgenic animals containing all genetic components: the optimised orthogonal synthetase/tRNA pair Smad4-NES::PCKRS/tRNA(C15), the PC-Cre construct, and the floxed Cre target ChR2::mKate2. All protein coding components were driven by a *glr-1p* promoter, an orthologue of human GRIA1 (glutamate ionotropic receptor AMPA type subunit 1). This promoter is active in glutamatergic neurons, including command interneurons (*Maricq et al., 1995*; *Figure 3—figure supplement 1A*). We grew animals on PCK from the L1 larval stage and activated PC-Cre by UV illumination when they had reached the L4 larval stage (after 2 days). 24 hr after activation, we saw strong expression of ChR2::mKate2 in neurons expressing PC-Cre. In contrast, we saw no expression of ChR2::mKate2 in animals which had undergone UV illumination without prior feeding on PCK, and in animals fed on PCK but which had not undergone UV illumination (*Figure 3—figure supplement 1B*). Furthermore, as expected for a membrane channel, the red fluorescence of ChR2::mKate2 was localised at the cellular membrane. We observed red fluorescence only in cells expressing the *glr-1p* promoter, as evidenced by overlap with *glr-1p*-driven expression of GFP.

Using the optimised Smad4-NES::PCKRS/tRNA(C15) expression system, we observed expression of the target locus, indicative of PC-Cre photoactivation, in 63 % of animals 24 hr after uncaging. We saw no red fluorescence without UV treatment. The 63 % activation rate was a vast improvement relative to the 1 % we observed with unmodified PCKRS and tRNA(Pyl) (*Figure 3D*).

## Improving photocaged Cre recombinase

To further improve the photoactivation method, we next turned to PC-Cre optimisation. When using amber stop codons for ncAA incorporation, the tRNA$_{CUA}$ competes with endogenous release factors. This competition results in a mixture of full-length polypeptides, with ncAA incorporated, as well as truncated polypeptides due to translational termination at the amber stop codon. As assessed with the GFP::mCherry incorporation reporter, we show that the percentage of full-length protein is between 4% and 5% when using the improved Smad4-NES::PCKRS/tRNA(C15) incorporation system (*Figure 2F*). It is likely that, similar to the fluorescent reporter, the majority of translation events of the PC-Cre mRNA will also terminate at the internal amber stop codon, even in the presence of PCK. A Cre protein truncated at the amber stop codon in position 201 is missing the majority of its active site, but the parts of Cre responsible for the protein-protein interaction required for active Cre tetramer formation are still present, as are large parts of the DNA binding interface (*Guo et al., 1997*). We initially based PC-Cre on previously reported Cre constructs, which contain an N-terminal SV40 NLS in addition to the internal NLS native to Cre (*Luo et al., 2016*; *Brown et al., 2018b*; *Le et al., 1999*; *Macosko et al., 2009*), thus ensuring localisation of the enzyme to the nucleus. We cannot exclude the possibility that truncated Cre, which locates to the nucleus due to the NLSs upstream of position 201, may interfere with tetramer formation or DNA binding.

We aimed to utilise the *C. elegans* nuclear import machinery to enrich for full-length product in the nucleus. Since Cre acts on nuclear DNA, we reasoned that we could boost PC-Cre activity

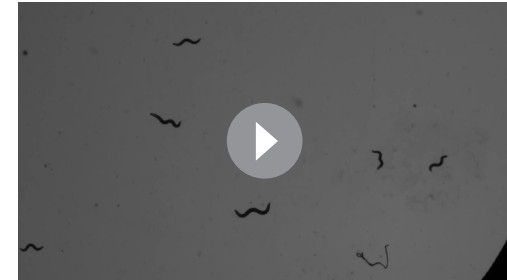

**Video 1.** Response to 470 nm light stimulation of animals expressing *glr-1p::optPC-Cre*, grown on photocaged lysine (PCK), uncaged, and then shifted to plates containing all-trans-retinal (ATR). Quantifications of the response of this line and a further, independent, line are shown in Figure 3G and Figure 3—figure supplement 1C.

https://elifesciences.org/articles/67075/figures#video1

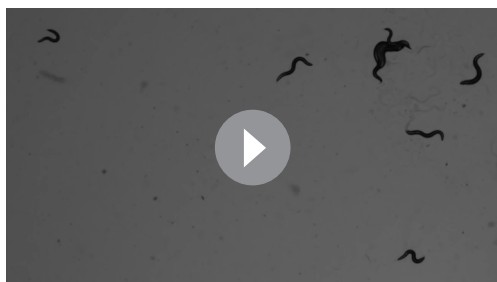

**Video 2.** Response to 470 nm light stimulation of animals expressing *glr-1p::optPC-Cre*, grown on photocaged lysine (PCK), uncaged, and then shifted to plates without all-trans-retinal (ATR). Quantifications of the response of this line and a further, independent, line are shown in Figure 3G and Figure 3—figure supplement 1C.

https://elifesciences.org/articles/67075/figures#video2

by lowering the fraction of truncated PC-Cre while increasing the fraction of full-length PC-Cre present in the nucleus. To this end, we removed both NLSs upstream of position 201 by introducing a R119A mutation disabling the internal NLS sequence (*Le et al., 1999*) and by removing the N-terminal SV40 NLS. To restore nuclear import for full-length PC-Cre only, we attached the strong *C. elegans* EGL-13 NLS (*Lyssenko et al., 2007*) to the Cre C-terminus to create optimised PC-Cre (optPC-Cre) (*Figure 3E*). When we compared optPC-Cre to PC-Cre, we indeed found that target gene activation was significantly improved: from 63 % for PC-Cre to 97 % for optPC-Cre (*Figure 3D*). We saw no expression of the ChR2::mKate2 target gene without PCK or UV activation (*Figure 3D and F*).

We next performed behavioural assays to assess whether our system allowed optical control of neurons. After activation of optPC-Cre, we waited 24 hr to achieve sufficient expression of ChR2::mKate2, which we confirmed by visual inspection under a dissecting fluorescence microscope. When we exposed worms to 470 nm blue light to activate ChR2, we observed clear reversals in worms expressing ChR2::mKate2 in the presence of the ChR2 cofactor all-trans-retinal (ATR). In animals expressing ChR2::mKate2 in the absence of ATR, we did not observe such reactions (*Figure 3G*, *Figure 3—figure supplement 1C*, *Video 1* and *Video 2*). The induced behaviour concurs with previous studies showing that optogenetic activation of neurons expressing ChR2 from the *glr-1p* promoter induces backward movement (*Schultheis et al., 2011*). When we compared strains expressing PC-Cre and optPC-Cre, respectively, we saw the strongest and most robust responses in the strains where optPC-Cre was used to control expression of ChR2::mKate2, followed by strains using non-optimised PC-Cre, but using the efficient Smad4-NES::PCKRS/tRNA(C15) incorporation system. Animals expressing unmodified PCKRS and non-optimised Cre showed almost no response to optogenetic activation (*Figure 3G*, *Figure 3—figure supplement 1C*). Taken together, we conclude that optPC-Cre is an effective tool with which to control optogenetic channel expression for the purpose of optical control of neuronal activity.

## Cell-specific gene activation using LaserTAC

The use of light to control activity of Cre recombinase should allow precise spatial control of Cre-dependent DNA recombination. We tested the precision of uncaging by using a microscope mounted 365 nm laser to target individual cells in the touch response circuitry. *C. elegans* has six mechanosensory neurons for the perception of soft touch: AVM, ALML, and ALMR, which are located in the anterior of the worm, and PVM, PLML, and PLMR, which are located in the posterior (*Figure 4A*). Activation of the touch receptor neurons results in avoidance behaviour. Worms respond with backward movement to an anterior touch, and with forward movement to a posterior touch (*Chalfie et al., 1985*). Previous studies using targeted ChR2 illumination in individual animals likewise showed that optogenetic activation of the anterior neurons AVM and ALM results in reversals, while activation of the posterior neurons PVM and PLM results in forward movement. Concurrent activation of all six neurons results in reversals in the majority of cases. Repeated activation leads to a reduced response due to habituation (*Leifer et al., 2011*; *Stirman et al., 2011*; *Schild and Glauser, 2015*).

We aimed to investigate the posterior PLM neurons by using LaserTAC to selectively express the optogenetic channel Chrimson (*Schild and Glauser, 2015*) in both PLM neurons, or in PLML and PLMR individually. To this end, we generated strains with Smad-4-NES::PCKRS and optPC-Cre expression driven by the *mec-7p* promoter, which is active in all six mechanosensory neurons (*Mitani et al., 1993*). To express Chrimson, we used a Chrimson::mKate2 fusion gene separated from the panneuronal *maco-1p* promoter by a transcriptional terminator flanked by loxP sites (*Figure 4—figure supplement 1A*). We chose the panneuronal *maco-1p* promoter, an orthologue of human MACO1

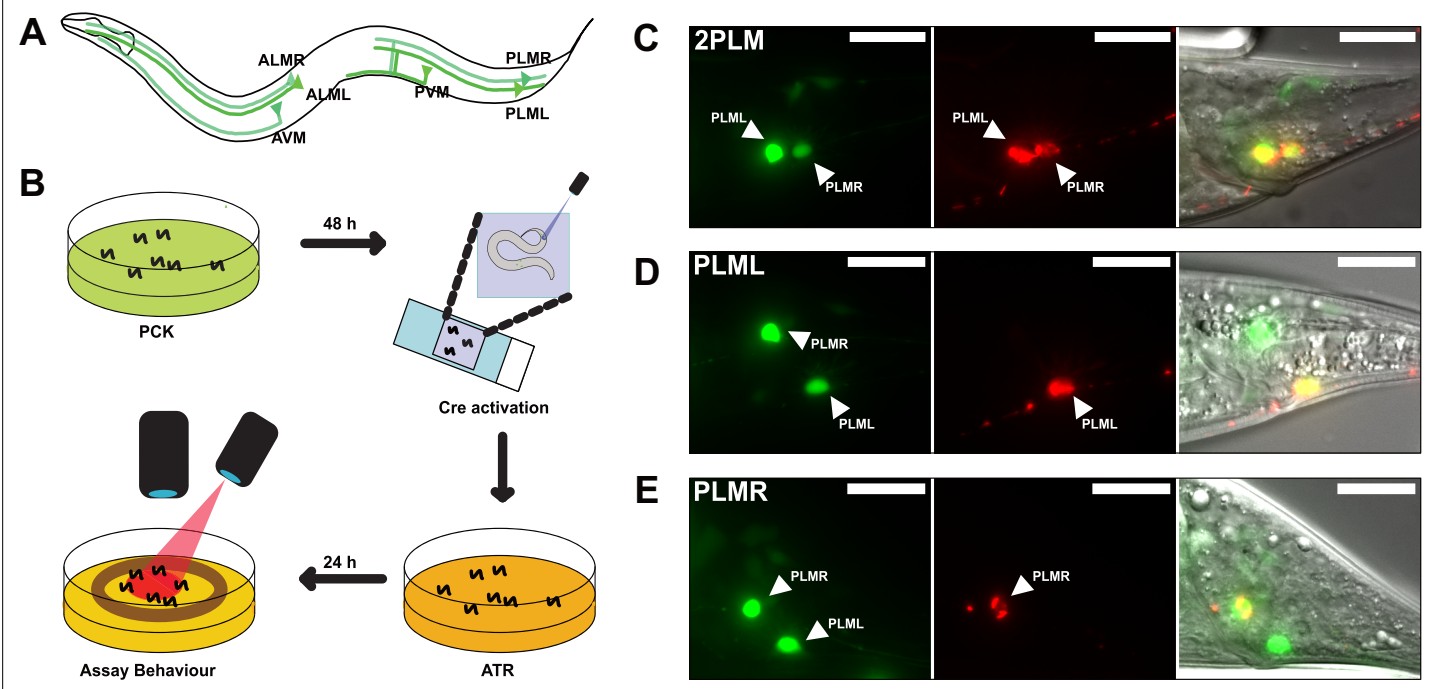

**Figure 4.** LaserTAC using optimised photocaged Cre-recombinase facilitates cell-specific expression of optogenetic channels in *C.elegans*. (**A**) Map of the six *C. elegans* touch receptor neurons showing the posterior location of the PLM neuron pair. (**B**) Diagram of experimental procedure for activating expression of target genes in single cells using LaserTAC, followed by optogenetic behavioural assay. (**C–E**) Fluorescent images showing targeted expression of Chrimson::mKate2 24 hr after laser induced activation of optPC-Cre (middle panels) in both PLM neurons (**C**), only in PLML (**D**), only in PLMR (**E**). Scale bars 20 µm. GFP signal in the left panels indicates cells expressing optPC-Cre.

The online version of this article includes the following figure supplement(s) for figure 4:

**Figure supplement 1.** Genetic constructs for expression of optPC-Cre in touch receptor neurons and soft touch assays to measure touch receptor neuron function.

**Figure supplement 1—source data 1.** Source data for soft touch assays shown in *Figure 4—figure supplement 1C,D*.

(macoilin 1) (*Arellano-Carbajal et al., 2011*), because in contrast to *mec-7p*, it shows increasing expression from the L4 larval to the adult stage, the age when we aimed to induce Chrimson::mKate2 expression (*Figure 4—figure supplement 1B*).

To activate expression of the Chrimson channel in the PLM neurons, we mounted age-synchronised L4 stage animals, grown on PCK from L1, on a microscope slide and used a Micropoint Galvo system to deliver a short 2–4 s 365 nm pulse to individual cells, targeting the nucleus. After uncaging, we transferred the worms to an NGM agar plate without PCK for recovery and to allow for induced Chrimson::mKate2 expression (*Figure 4B*). After 12–24 hr, we observed Chrimson::mKate2 expression that was only present in laser-targeted cells. We were able to selectively activate optPC-Cre in both PLM neurons, as well as in PLML or PLMR alone (*Figure 4C–E*).

We confirmed that the presence of the PCK incorporation machinery and optPC-Cre did not affect the function of the touch sensory neurons by assaying the animals response to soft touch (*Chalfie et al., 2014*; *Figure 4—figure supplement 1C,D*).

## Optogenetic manipulation of PLM touch sensory neurons reveals distinct and synergistic roles for PLML and PLMR in the soft touch response

To determine the behavioural response to PLM stimulation, we prepared animals expressing Chrimson::mKate2 in either both PLM neurons, or in PLML or PLMR alone. After optPC-Cre activation, we grew the worms on plates supplemented with ATR for 24 hr to allow expression of the optogenetic channel and confirmed the desired expression pattern under a fluorescence dissecting microscope.

We then activated Chrimson::mKate2 expressing neurons by illumination for 1 s every 30 s with 617 nm at the maximum power setting of 74 mW/cm$^2$. During the assay, the animals were moving freely on the plate and we illuminated the entire plate. We found that illumination of animals expressing Chrimson::mKate2 in both PLM neurons ('2PLM') triggered a robust response with the worms initiating forward movement (*Figure 5A,B*, *Figure 5—figure supplement 1*, *Video 3*). Conversely, mock-treated animals ('0PLM') grown on PCK and ATR but without activation of optPC-Cre showed no response (*Figure 5—figure supplement 1*, *Video 4*). Animals expressing Chrimson::mKate2 only in the PLML or the PLMR neurons respectively also showed a clear reaction to stimulation (*Figure 5A,B*, *Figure 5—figure supplement 1*, *Video 5* and *Video 6*). For all three expression patterns (2PLM, PLML, PLMR), we observed a gradual reduction in the response with increasing number of stimulations, both in the fraction of animals responding and in the speed upon stimulation, consistent with habituation to the stimulus (*Stirman et al., 2011*). We found that the 2PLM animals responded stronger to stimulation than animals expressing Chrimson::mKate2 in either PLML or PLMR alone, both in regard to the fraction of animals responding, and by velocity (*Figure 5A,B*).

We then proceeded to more closely investigate the responses to activation of PLML and PLMR individually. The members of the PLM neuron pair, PLML and PLMR, show a striking asymmetry in their connections within the worm connectome (*Figure 5C*; *Chalfie et al., 1985*). PLMR forms chemical synapses connecting it to 10 downstream neurons including the AVA and AVD interneuron pairs, which are involved in driving backward locomotion. In addition to chemical synapses, PLMR also forms gap junctions to PVCR, PHCR, and LUAR. PLML, on the other hand, is connected to the touch response circuit exclusively through gap junctions, formed with the interneurons PVCL, PHCL, and LUAL. In fact, PLML does not form any chemical synapses, with the exception of a single connection to HSNL, a neuron involved in the control of egg laying. Asymmetric wiring of PLML and PLMR has been proposed to influence the *C. elegans* tap response, a withdrawal reflex that integrates information from the anterior and posterior mechanosensory neurons. In worms with ablation of both PLM neurons and the single PLMR neuron, a tap stimulus increased backward movement. In contrast, ablating PLML had no effect (*Wicks and Rankin, 1995*).

As we saw no difference between PLML and PLMR in the response to strong stimulation (*Figure 5A,B*), we decided to test the animals' responses to weaker stimulation of the PLM neurons. For this, we reduced the duration of the 617 nm light pulse from 1 s to 0.1 s and delivered varying low intensities, starting at 3 mW/cm$^2$, the lowest setting that was possible with our setup. While the 0.1 s, 3 mW/cm$^2$ stimulus was sufficient to elicit a response in animals expressing Chrimson in both PLMs, neither PLML nor PLMR animals reacted, indicating a synergistic effect of PLML and PLMR and the requirement for activation of both neurons to elicit a reaction in response to weak stimuli (*Figure 5D–F*, *Figure 5—figure supplement 2*).

When we increased illumination intensity, we saw a striking difference appear between the responses to PLML and PLMR stimulation, respectively. Animals expressing Chrimson in PLMR showed a robust reaction to stimulation at 14 mW/cm$^2$ and higher. In contrast, the response of PLML animals was consistently below that of PLMR or 2PLM animals (*Figure 5D–F*, *Figure 5—figure supplement 2*). Only at the highest intensity setting of 74 mW/cm$^2$ did we observe animals of all three expression patterns responding robustly to the stimulus. The observed differences in response for PLML and PLMR are not due to differential expression of Chrimson::mKate2 (*Figure 5—figure supplement 3*), therefore they are likely due to the asymmetric connectivity of the two neurons within the *C. elegans* nervous system.

## Discussion

We demonstrate a robust, improved method for highly efficient ncAA incorporation in *C. elegans*. Our optimised ncAA incorporation system will provide the means to introduce a wide range of new chemical functionalities into proteins, such as crosslinkers, bioorthogonal groups, and post-translational modifications, and serve as a catalyst for new approaches to manipulate and understand cellular biology in *C. elegans* (*Chin, 2017*; *Davis and Chin, 2012*; *Nguyen et al., 2018*; *Lang and Chin, 2014*; *Young and Schultz, 2018*; *Zhang et al., 2015*; *Baker and Deiters, 2014*). Furthermore, since our optimisation approach is based on components that are not specific to *C. elegans*, we expect that it will be easily exportable to other organisms.

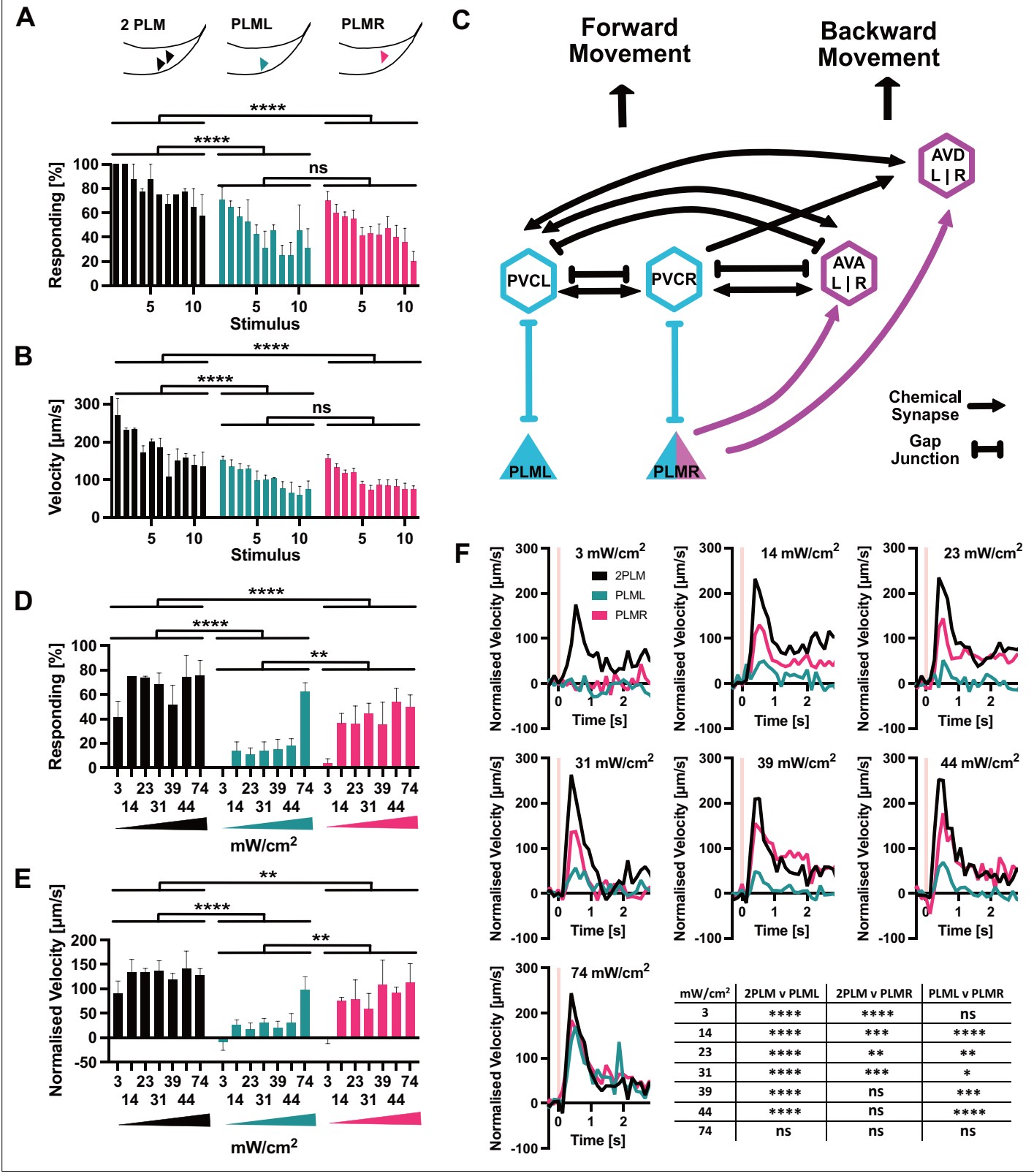

**Figure 5.** PLM neurons show distinct responses to optogenetic stimulation. (**A**) Percentage of animals responding to repeated stimulations with light pulses of 1 s, 74 mW/cm², 617 nm, delivered every 30 s. Worms expressed Chrimson in either both PLM neurons ('2PLM', black), only in PLML (blue-green), or only in PLMR (red). Responding worms are defined as those having mean forward speeds > 100 µm s⁻¹ over the 2 s following stimulation. Assays were performed with cohorts of 4–6 worms stimulated simultaneously, one cohort constitutes a replicate. Replicates were assayed on different

*Figure 5 continued on next page*

*Figure 5 continued*

days. Significances derived from two-way ANOVA (2PLM n = 2; PLML and PLMR n = 4). (**B**) Mean velocity for the experiments analysed in (**A**). Mean velocity was determined for each cohort. Significances derived from two-way ANOVA (2PLM n = 2; PLML and PLMR n = 4). (**C**) Simplified schematic of the asymmetric neuronal connectivity of PLML and PLMR within the touch circuit. Symmetric PLM connections are shown in light blue, whereas asymmetric connections shown in magenta. (**D**) Percentages of animals which respond to stimulation (0.1 s, 617 nm) at the power levels indicated. Chrimson expressed in either both PLM neurons ('2PLM', black), only in PLML (blue-green) or only in PLMR (red). For each individual worm, the velocity was measured in the 1 s window from 0.4 s to 1.2 s after stimulation, which corresponds to the peak response. Velocities were then normalised by subtracting the average speed of the individual over the 0.5 s immediately prior to stimulation. Responding worms are defined as those having mean normalised forward speeds >100 μm/s in the window from 0.4 s to 1.2 s after stimulation. Cohorts of 4–8 animals were stimulated together, and the mean velocity of one cohort constitutes a replicate. Replicates were assayed on different days. Significances derived from two-way ANOVA, n = 3 for all conditions. (**E**) Mean normalised velocity for the experiments analysed in (**D**). The mean normalised velocity across all animals within a cohort was determined for the 1 s window 0.2–1.2 s after stimulation. Significances derived from two-way ANOVA. n = 3 for all conditions. (**F**) Normalised mean velocity traces of the experiments analysed for (**D,E**) across all three replicates for each condition at the power levels indicated. 2PLM (black), PLML (blue-green), and PLMR (red). Pink vertical lines indicate the light pulse. Statistical significances are provided in the table in (**F**). Significance was determined by two-way ANOVA using the mean of the three replicates for each of the seven speed measurements between 0.4 s and 1.2 s after stimulation, which corresponds to the peak response. *p<0.05, **p<0.01, ***p<0.001, ****p<0.0001. Traces before normalisation are shown in *Figure 5—figure supplement 2*.

The online version of this article includes the following figure supplement(s) for figure 5:

**Source data 1.** Velocities for all individuals and experiments depicted in *Figure 5A,B* and *Figure 5—figure supplement 1*.

**Source data 2.** Velocities for all individuals in experiments depicted in *Figure 5C–E* and *Figure 5—figure supplement 2*.

**Figure supplement 1.** Speed traces used in *Figure 5A,B*.

**Figure supplement 2.** Traces before normalisation of experiments shown in *Figure 5D–F*.

**Figure supplement 3.** Expression levels of Chrimson::mKate2.

**Figure supplement 3—source data 1.** Fluorescence intensities measured in PLML and PLMR.

The genetic code expansion approach relies on the use of an amber stop codon to direct incorporation of ncAA, which means that endogenous TAG codons may be inadvertently suppressed by the orthogonal system, albeit likely at low levels. Interestingly, *C. elegans* and other organisms have evolved natural surveillance mechanisms to mitigate against the read-through of stop codons (*Arribere et al., 2016*), which will also mitigate against any potential effects caused by our system. Indeed, we find that the function of the mechanosensory neurons we assay is unaffected by the presence of the orthogonal ncAA incorporation machinery.

We demonstrate the utility of our system by creating LaserTAC, which is based on an optimised photoactivatable Cre recombinase and offers several unique advantages for optical control of gene expression. First, the recombinase activity is tightly controlled; the presence of the photocaging group completely blocks activity and upon photoactivation, highly efficient wildtype Cre is generated. Second, following photoactivation, the photocaged amino acid required for amber stop codon suppression can be removed. Thus translation of full-length Cre ceases, leaving worms devoid of photocaged Cre for subsequent biological studies (*Maywood et al., 2018*). Third, due to the stability

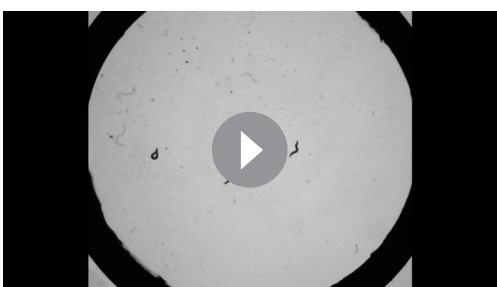

**Video 3.** Response to 617 nm light stimulation of animals expressing Chrimson::mKate2 in both PLM neurons and grown in the presence of all-trans-retinal (ATR).

https://elifesciences.org/articles/67075/figures#video3

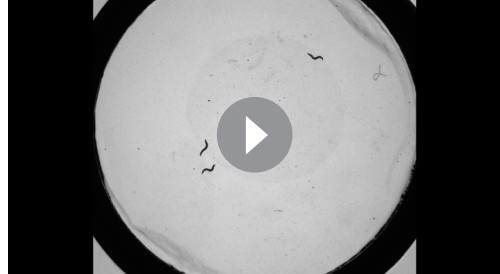

**Video 4.** Response to 617 nm light stimulation of mock treated animals, grown on photocaged lysine (PCK) and in the presence of all-trans-retinal (ATR), but without uncaging.

https://elifesciences.org/articles/67075/figures#video4

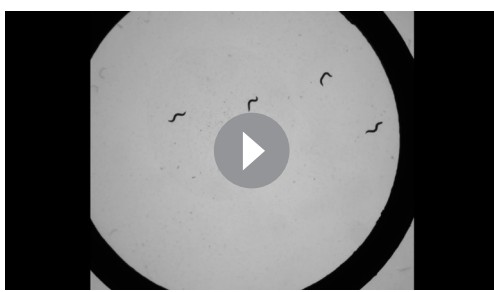

**Video 5.** Response to 617 nm light stimulation of animals expressing Chrimson::mKate2 only in PLML and grown in the presence of all-trans-retinal (ATR). https://elifesciences.org/articles/67075/figures#video5

of the photocaging group at longer wavelengths, our photoactivatable Cre is fully compatible with imaging and optogenetic methods that use visible light.

We used LaserTAC to switch on optogenetic channel expression in individual neurons within a bilaterally symmetric pair that cannot be targeted by genetic means. Switching on gene expression using LaserTAC is fast and easily enables the preparation of dozens of animals for downstream experiments. Light activation requires only a microscope-mounted laser, generally used for ablation in most *C. elegans* laboratories, modified to emit light at 365 nm. Generating the required transgenic strains is straightforward and at present involves the introduction of the genetic constructs, followed by genomic integration. The power of the method we present here lies in the versatility it offers, not only in respect to the ability to investigate single neurons. Once a transgenic strain is made, any combination of neurons targeted by the promoter that was used can in principle be investigated. This removes the requirement for the creation of a potentially large numbers of lines by allowing multiple conditions to be investigated using a single strain. A further advantage offered by our approach is the ability of precise temporal control, which is generally not available using promoter-based systems.

We demonstrate the capabilities of LaserTAC by targeting expression of the optogenetic channel Chrimson to the mechanosensory PLM neurons. This enables us to analyse the individual contributions of the neurons PLML and PLMR to the *C. elegans* tail touch circuit response. Since we target expression of the optogenetic channel, we can perform behavioural assays by globally illuminating a plate of freely moving animals.

We found that PLML and PLMR show some redundancy in triggering the tail touch response after strong stimulation. However, using weaker stimuli, we found that only stimulation of both neurons elicits a response, indicative of a synergistic rather than a simply redundant role for PLML and PLMR. Interestingly, our results demonstrate asymmetric contributions of PLML and PLMR to the touch response. Considering the electrical and chemical connectome of this circuit (*Figure 5C*), our results are consistent with a model where the touch response is controlled by two parallel pathways that rely on gap junctions or chemical synapses, respectively. The first pathway involves direct connections of both PLM neurons to the PVC interneurons via gap junctions. Activation of the PVCs through these gap junctions triggers forward motion. The PVCs form connections with forward command interneuron AVB as well as inhibitory connections with backward command interneurons AVA/AVD. The second pathway involves chemical synaptic transmission from PLMR onto AVA and AVD, inhibiting backward locomotion. Thus, when only PLMR is stimulated, both pathways are activated, resulting in a stronger response, whereas when PLML alone is stimulated only the gap junction pathway is activated, resulting in a weaker response. Unlike the PLMs, the PVCs are connected to each other by gap junctions and may thus act as electrical sinks for signals from the PLMs. This could help to explain the requirement for stronger stimulation of the single neurons, especially for PLML, to trigger a response as compared to the stimulation of both PLMs simultaneously. Stimulation of both PLM neurons together results in full activation of the gap junction pathway by activating both PVC neurons, which, together with the direct inhibition of AVA/AVD by PLMR, results in full activation of the tail touch motor response. The strong

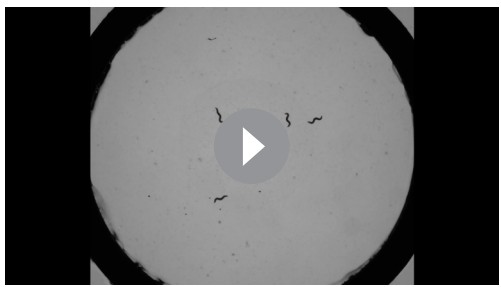

**Video 6.** Response to 617 nm light stimulation of animals expressing Chrimson::mKate2 only in PLMR and grown in the presence of all-trans-retinal (ATR). https://elifesciences.org/articles/67075/figures#video6

synergistic effect we observe with the simultaneous stimulation of PLML and PLMR indicates a requirement for the stimulation of both PVC neurons.

While we focused here on expression of optogenetic channels in *C. elegans* neurons, simple extensions of our method will enable the cell-specific expression of other desired transgenes of interest. By inserting loxP sites into genomic loci using CRISPR/Cas9, it should be possible to control, at single-cell resolution, the expression of any endogenous protein, such as neuropeptides, receptors, innexins, proteins involved in synaptic transmission, etc. Since the components we employ are functional in other *C. elegans* tissues (*Greiss and Chin, 2011*) and indeed in other model systems such as zebrafish, fruit fly, mouse, cultured cells, and ex vivo tissues, we anticipate that our method will have broad applicability beyond the nervous system and beyond *C. elegans*.

# Materials and methods

**Key resources table**

| Reagent type (species) or resource | Designation | Source or reference | Identifiers | Additional information |
|---|---|---|---|---|
| Genetic reagent (*Caenorhabditis elegans*) | SGR30 | This paper | n/a | *greEx17[sur-5p::FLAG::PCKRS::GFP]*<br>***Figure 2D*, *Figure 2—figure supplement 1B*** |
| Genetic reagent (*C. elegans*) | SGR31 | This paper | n/a | *greEx18[sur-5p::p120cts::PCKRS::GFP]*<br>***Figure 2D*, *Figure 2—figure supplement 1B*** |
| Genetic reagent (*C. elegans*) | SGR32 | This paper | n/a | *greEx19[sur-5p::PKIα::PCKRS::GFP]*<br>***Figure 2D*, *Figure 2—figure supplement 1B*** |
| Genetic reagent (*C. elegans*) | SGR33 | This paper | n/a | *greEx20[sur-5p::smad-4::PCKRS::GFP]*<br>***Figure 2D*, *Figure 2—figure supplement 1B*** |
| Genetic reagent (*C. elegans*) | SGR34 | This paper | n/a | *greEx21[sur-5p::S-NES::PCKRS::GFP]*<br>***Figure 2D*, *Figure 2—figure supplement 1B*** |
| Genetic reagent (*C. elegans*) | SGR35 | This paper | n/a | *greEx22[sur-5p::PCKRS; rpr-1p::tRNA(Pyl); rps-0p::GFP(am)::mCherry::HA]; smg-2(e2008)*<br>***Figure 2C, E, F and G*, *Figure 2—figure supplement 1A*** |
| Genetic reagent (*C. elegans*) | SGR36 | This paper | n/a | *greEx23[sur-5p::p120cts::PCKRS; rpr-1p::tRNA(Pyl); rps-0p::GFP(am)::mCherry::HA];smg-2(e2008)*<br>***Figure 2C*, *Figure 2—figure supplement 1A*** |
| Genetic reagent (*C. elegans*) | SGR37 | This paper | n/a | *greEx24[sur-5p::PKIα::PCKRS; rpr-1p::tRNA(Pyl); rps-0p::GFP(am)::mCherry::HA]; smg-2(e2008)*<br>***Figure 2C*, *Figure 2—figure supplement 1A*** |
| Genetic reagent (*C. elegans*) | SGR38 | This paper | n/a | *greEx25[sur-5p::smad-4::PCKRS; rpr-1p::tRNA(Pyl); rps-0p::GFP(am)::mCherry::HA]; smg-2(e2008)*<br>***Figure 2C*, *Figure 2—figure supplement 1A*** |
| Genetic reagent (*C. elegans*) | SGR39 | This paper | n/a | *greEx26[sur-5p::SNES::PCKRS; rpr-1p::tRNA(Pyl); rps-0p::GFP(am)::mCherry::HA]; smg-2(e2008)*<br>***Figure 2C*, *Figure 2—figure supplement 1A*** |
| Genetic reagent (*C. elegans*) | SGR40 | This paper | n/a | *greEx27[sur-5p::PCKRS; rpr-1p::tRNA(Pyl); rps-0p::GFP(am)::mCherry::HA]; smg-2(e2008)*<br>***Figure 2F,G*** |
| Genetic reagent (*C. elegans*) | SGR45 | This paper | n/a | *greEx32[sur-5p::PKIα::PCKRS; rpr-1p::tRNA(C15); rps-0p::GFP(am)::mCherry::HA]; smg-2(e2008)*<br>***Figure 2C, E, F and G*, *Figure 2—figure supplement 1A,C*** |
| Genetic reagent (*C. elegans*) | SGR46 | This paper | n/a | *greEx33[sur-5p::smad-4::PCKRS; rpr-1p::tRNA(C15); rps-0p::GFP(am)::mCherry::HA]; smg-2(e2008)*<br>***Figure 2C, E, F and G*, *Figure 2—figure supplement 1A,C*** |
| Genetic reagent (*C. elegans*) | SGR47 | This paper | n/a | *greEx34[sur-5p:: PKIα::PCKRS; rpr-1p::tRNA(C15); rps-0p::GFP(am)::mCherry::HA];smg-2(e2008)*<br>***Figure 2F,G*** |

*Continued on next page*

*Continued*

| Reagent type (species) or resource | Designation | Source or reference | Identifiers | Additional information |
|---|---|---|---|---|
| Genetic reagent (*C. elegans*) | SGR48 | This paper | n/a | *greEx35[sur-5p::smad-4::PCKRS; rpr-1p::tRNA(C15); rps-0p::GFP(am)::mCherry::HA];smg-2(e2008)* **Figure 2F,G** |
| Genetic reagent (*C. elegans*) | SGR49 | This paper | n/a | *greEx36[glr-1p::PCKRS rpr-1p::tRNA(Pyl); glr-1p::PC-Cre; glr-1p::B-gal terminator+ loxP::ChR2::mKate2]* **Figure 3D,G, Figure 3—figure supplement 1C** |
| Genetic reagent (*C. elegans*) | SGR50 | This paper | n/a | *greEx37[glr-1p::PCKRS; rpr-1p::tRNA(Pyl); glr-1p::PC-Cre; glr-1p::B-gal terminator+ loxP::ChR2::mKate2]* **Figure 3D,G, Figure 3—figure supplement 1C** |
| Genetic reagent (*C. elegans*) | SGR51 | This paper | n/a | *greEx38[glr-1p::smad-4::PCKRS; rpr-1p::tRNA(C15); glr-1p::PC-Cre; glr-1p::B-gal terminator+ loxP::ChR2::mKate2]* **Figure 3D,G, Figure 3—figure supplement 1B,C** |
| Genetic reagent (*C. elegans*) | SGR52 | This paper | n/a | *greEx39[glr-1p::smad-4::PCKRS; rpr-1p::tRNA(C15); glr-1p::PC-Cre; glr-1p::B-gal terminator+ loxP::ChR2::mKate2]* **Figure 3D,G, Figure 3—figure supplement 1C** |
| Genetic reagent (*C. elegans*) | SGR53 | This paper | n/a | *greEx40[glr-1p::smad-4::PCKRS; rpr-1p::tRNA(C15); glr-1p::optPC-Cre; glr-1p::B-gal terminator+ loxP::ChR2::mKate2]* **Figure 3D, F and G, Figure 3—figure supplement 1C , Video 1, Video 2** |
| Genetic reagent (*C. elegans*) | SGR54 | This paper | n/a | *greEx41[glr-1p::smad-4::PCKRS; rpr-1p::tRNA(C15); glr-1p::optPC-Cre; glr-1p::B-gal terminator+ loxP::ChR2::mKate2]* **Figure 3D,G, Figure 3—figure supplement 1C** |
| Genetic reagent (*C. elegans*) | SGR55 | This paper | n/a | *greEx42[mec-7p::smad-4::PCKRS; rpr-1p::tRNA(C15); mec-7p::optPC-Cre; Pmaco-1::B-gal terminator+ loxP::ChR2::mKate2]* |
| Genetic reagent (*C. elegans*) | SGR56 | This paper | n/a | *greIs1[mec-7p::smad-4::PCKRS; rpr-1p::tRNA(C15); mec-7p::optPC-Cre; Pmaco-1::terminator+ loxP::ChR2::mKate2]* **Figure 4C–E** generated by gamma integration from SGR55, backcrossed 2× |
| Genetic reagent (*C. elegans*) | SGR96 | This paper | n/a | *greIs1[mec-7p::smad-4::PCKRS; rpr-1p::tRNA(C15); mec-7p::optPC-Cre; Pmaco-1::terminator+ loxP::ChR2::mKate2]* **Figure 4—figure supplement 1C,D, Figure 5, Figure 5—figure supplements 1–3** SGR56 backcrossed 2x |
| Genetic reagent (*C. elegans*) | SGR98 | This paper | n/a | *greEx18[sur-5p::p120cts::PCKRS::GFP]* **Figure 2—figure supplement 1B** |
| Genetic reagent (*C. elegans*) | SGR99 | This paper | n/a | *greEx19[sur-5p::PKIα::PCKRS::GFP]* **Figure 2—figure supplement 1B** |
| Genetic reagent (*C. elegans*) | SGR100 | This paper | n/a | *greEx20[sur-5p::smad-4::PCKRS::GFP]* **Figure 2—figure supplement 1B** |
| Genetic reagent (*C. elegans*) | SGR101 | This paper | n/a | *greEx21[sur-5p::S-NES::PCKRS::GFP]* **Figure 2—figure supplement 1B** |
| Strain, strain background (*C. elegans*) | N2 | CGC | WBStrain00000001 | Wild type **Figure 4—figure supplement 1C** |
| Genetic reagent (*C. elegans*) | CZ10175 | CGC | WBStrain00005421 | *zdIs5[mec-4p::GFP+ lin-15(+)]* **Figure 4—figure supplement 1C** ("*mec-4p::gfp*") |

*Continued*

| Reagent type (species) or resource | Designation | Source or reference | Identifiers | Additional information |
|---|---|---|---|---|
| Genetic reagent (*C. elegans*) | | Chalfie and Au 1989 (10.1126/ science.2646709) | WBVar00266589 | *mec-4(u231)* *Figure 4—figure supplement 1C ("mec-4(d)")* |
| Antibody | Mouse anti-GFP, monoclonal (clones 7.1 and 13.1) | Roche | Cat# 11814460001 RRID:AB_390913 | *Figure 2C, F and G*, *Figure 2—figure supplement 1A* (1:4000) |
| Antibody | Rat anti-HA, monoclonal (clone 3F10) | Roche | Cat# 11867423001 RRID:AB_390918 | *Figure 2C,G* (1:2000) |
| Antibody | Horse anti-mouse IgG HRP | Cell Signaling Technology | Cat# 7076 S RRID:AB_330924 | *Figure 2C, F and G*, *Figure 2—figure supplement 1A* (1:5000) |
| Antibody | Goat anti-Rat IgG (H + L) HRP | Thermo Fisher Scientific | Cat# 31470 RRID:AB_228356 | *Figure 2C,G* (1:5000) |
| Chemical compound | Photocaged lysine (6-nitropiperonyl-L-Lysine) | ChiroBlock GmbH; *Gautier et al., 2010* | CAS number: 1221189-11-2 | Custom synthesised by ChiroBlock GmbH, Germany. Synthesis described in Gautier et al. (10.1021/ ja910688s) |

## Plasmids

All expression plasmids were generated from pENTR plasmids using the Gateway system (Thermo Fisher Scientific). All plasmids are described in *Supplementary file 1*.

## *C. elegans* strains

Strains were maintained under standard conditions unless otherwise indicated (*Brenner, 1974*; *Stiernagle, 2006*).

Transgenic worms were generated by biolistic bombardment using hygromycin B as a selection marker (*Greiss and Chin, 2011*; *Radman et al., 2013*; *Davis and Greiss, 2018*) in either N2 or *smg-2(e2008)* genetic background. The *smg-2(e2008)* background lacks a functional nonsense-mediated decay machinery and was used to increase levels of reporter mRNA (*Greiss and Chin, 2011*). Gamma-irradiation to generate the integrated line SGR56 from SGR55 was carried out by Michael Fasseas (Invermis/Magnitude Biosciences). After integration, the line was backcrossed into N2 and subsequently maintained on standard NGM without added hygromycinB. SGR56 was backcrossed twice, SGR96 was backcrossed four times. All non-integrated lines were maintained on NGM supplemented with hygromycin B (0.3 mg/ml; Formedium). Strains used in this paper are listed in the key resources table.

## Imaging and image analysis

All imaging was carried out on a Zeiss M2 imager. Worms were mounted on glass slides for imaging and immobilised in a drop of M9 supplemented with 25 mM $NaN_3$.

Fluorescent images were analysed using ImageJ software. To determine nuclear to cytoplasmic ratios of NES::PCKRS::GFP variants, mean fluorescence intensity was measured for a region of interest within the nucleus and divided by the mean intensity of a region in the cytoplasm proximal to the nucleus. To compare Chrimson::mKate2 levels in the PLML and PLMR neurons, the threshold function was used to create regions of interest encompassing the neurons and the mean intensity of those regions was taken.

## ncAA feeding

PCK (*Gautier et al., 2010*) was custom synthesised by ChiroBlock GmbH, Germany. PCK-NGM plates were prepared by dissolving PCK powder in a small volume of 0.02 M HCl and adding the solution to molten NGM. The HCl in the NGM was neutralised by addition of equimolar amounts of NaOH as previously described (*Davis and Greiss, 2018*).

Animals were age synchronised by bleaching (*Stiernagle, 2006*) and added to PCK-NGM plates as L1 larvae. Food was then added to the PCK-NGM plates in the form of solubilised freeze-dried OP50 (LabTIE) reconstituted according to the manufacturer's instructions.

## Worm lysis and western blotting

Synchronised populations were grown on PCK-NGM plates until the young adult stage and washed off plates using M9 buffer supplemented with 0.001 % Triton-X100. Worms were settled, supernatant was removed, and worms were resuspended in undiluted 4× LDS loading buffer (Thermo Fisher Scientific) supplemented with NuPAGE Sample Reducing Agent (Thermo Fisher Scientific). Lysis was performed by a freeze/thaw cycle followed by 10 min incubation at 95 °C.

Samples were run on precast Bolt 4–12% gels (Thermo Fisher Scientific) for 19 min at 200 V. Proteins were transferred from the gel onto a nitrocellulose membrane using an iBlot2 device (Thermo Fisher Scientific).

After transfer, the membrane was blocked using 5 % milk powder in PBST (PBS + 0.1 % Tween-20) for 1 hr at room temperature. Incubation with primary antibodies was carried out in PBST + 5 % milk powder at 4 °C overnight. Blots were washed 6 × 5 min with PBST + 5 % milk powder before incubating with secondary antibody for 1 hr at room temperature.

Antibodies used are listed in the key resources table. Pierce ECL Western Blotting Substrate (Thermo Fisher Scientific) or SuperSignal West Femto chemiluminescent Substrate (Thermo Fisher Scientific) were used as detection agent. For quantitative blots, a C-DiGit Blot Scanner (LI-COR) was used, and intensities were analysed using ImageStudio software.

## Global uncaging of PCK for Cre activation

Worms were age synchronised and grown on PCK-NGM plates for 48 hr, washed onto unseeded 6 cm NGM plates, and illuminated for 5 min, 5 mW/cm$^2$ in a 365 nm CL-1000L crosslinker (UVP) as previously described (*Davis and Greiss, 2018*). After uncaging, worms were transferred to seeded NGM plates and scored for expression of the target gene 24 hr later by counting the animals showing expression of the red fluorescent marker. 400 µM FUDR was added to plates after uncaging to prevent hatching of F1 progeny and thus aid scoring of animals expressing the target gene. All plates were scored twice, independently by two people. Scoring was performed blind. Experiments were performed three times, each with two independent transgenic lines. Significance tests were carried out using Welch's t test as a pairwise comparison between each condition at each concentration using GraphPad Prism 8 software.

## Uncaging for Cre activation in single neurons

Worms were grown on 4 mM PCK from the L1 stage. For uncaging, worms were mounted on a 3 % agar and immobilised with 25 mM NaN$_3$ in M9 buffer. Targeting was performed on a Zeiss M2 Imager using a Micropoint Galvo module (Andor Technology/Oxford Instruments), fitted with a 365 nm dye cell. Neurons were identified using GFP as a guide, which was co-expressed with photocaged Cre, and the nucleus targeted with the laser. Each region was swept thrice using 10 repeat firing. The manual attenuator was set to full power and the Andor software attenuator to a power of 34 (we chose the power setting so that partial bleaching of GFP could be observed during Micropoint firing). After uncaging, the coverslip was removed and worms were washed off the pad onto seeded 6 cm plates using M9 + 0.001 % Triton-X. Mock laser treatment control worms were mounted and recovered similar to experimental worms but without exposure to the Micropoint laser.

## Behavioural assays

Immediately after uncaging, worms were transferred to NGM plates supplemented with ATR (30 µl 5 mM ATR dissolved in ethanol were added to the lawn of a seeded 6 cm NGM plate) or control NGM-only plates. Worms were left to recover on plates for 24 hr after uncaging.

Plates for behavioural assays were made approximately 1 hr before use. Fresh 6 cm NGM plates were seeded with a 20 µl drop of dilute OP50, prepared from freeze-dried OP50 (LabTIE). For this, the freeze-dried bacteria were reconstituted according to the manufacturer's instructions and a 40× dilution of the reconstituted bacteria was used to spot the plates. The lawn was then surrounded by a copper ring (inner diameter 17 mm) to prevent worms from leaving the camera field of view.

For experiments on PLM, expression of the target gene after optPC-Cre laser activation was visually confirmed using a Leica M165FC fluorescence microscope fitted with a 2.0× objective and animals showing the correct pattern (either both neurons or individual neurons, depending on the uncaging experiment) were picked onto the behaviour plates. For experiments using the *glr-1p* promoter, animals were randomly picked. After transfer, worms were left to acclimatise on behaviour plates for at least 10 min. Behavioural assays were carried out using a WormLab system (MBF Bioscience). Chrimson or Channelrhodopsin2 were activated using either the integrated 617 nm LED or 470 nm LED of the WormLab platform at full power (74 mW/cm$^2$ or 150 mW/cm$^2$, respectively), unless otherwise stated.

Tracking of worm behaviour was carried out using WormLab software (MBF Bioscience). For each worm, tracks detected by the WormLab software were consolidated into a single track. Data points where individual animals were not clearly identified by the software were discarded.

For all behavioural experiments, worms were grouped into cohorts. A cohort consisted of 4–10 animals undergoing the same treatment simultaneously on the same plate.

Behavioural experiments on worms expressing ChR2 from the *glr-1p* promoter were carried out by exposing worms to 1 s of 470 nm illumination at 150 mW/cm$^2$. Reversals were counted manually. The reversal percentages were determined for each replicate. Significance between conditions was tested using a Mann–Whitney U test.

Experiments using repeated stimulation of animals expressing Chrimson::mKate2 in the PLM neurons were carried out by exposing worms to repeated 1 s long stimulations with 617 nm light (74 mW/cm$^2$) at 30 s intervals. Raw speed data was binned and averaged at 1 s intervals for each animal. Worms were grouped by cohort. To determine the fraction of animals responding to a stimulus, worms with an average speed ≥100 µm s$^{-1}$ in the 2 s following stimulation were scored as responders. The 100 µm s$^{-1}$ threshold was selected as it constitutes the standard deviation of the mean speed of unstimulated worms. For each timepoint, the percentage of worms responding was determined for each cohort, constituting one replicate. Comparison between conditions was carried out by two-way ANOVA across all stimuli.

Experiments comparing conditions across different power levels for animals expressing Chrimson::mKate2 in the PLM neurons were carried out by subjecting each cohort of worms to 0.1 s stimulation with 617 nm at the indicated power levels. The percentage of worms responding for each cohort was calculated after first normalising the data by subtracting the mean speed for each individual worm in the 0.5 s immediately prior to stimulation. Then the mean velocity for each individual worm was determined for the response peak between 0.2 s and 1.2 s after stimulation. Worms were scored as responders if the mean normalised velocity was ≥ 100 µm s$^{-1}$.

When calculating the mean normalised velocity across experiments, the mean velocity among all animals within a cohort was determined first, followed by calculating the average of the mean velocities across cohorts. Significance for both analyses (fraction responding/intensity and mean velocity/intensity) was determined by two-way ANOVA.

## Analysis and statistics

All statistical analysis of data was carried out in GraphPad Prism 8. Statistical tests are stated in the figure legends.

All values of n in this article, unless otherwise stated, refer to biological replicates. For behavioural experiments, we define as biological replicates cohorts of worms that were assayed on different days. A cohort of worms consisted of animals undergoing treatment and assayed together on the same plate.

## Acknowledgements

We thank Maria Doitsidou, Netta Cohen, Kathrin Lang, Emanuel Busch, and Bill Schafer for helpful suggestions and input on the manuscript, and Zoltan Soltesz for help with initial experiments. Some strains were provided by the *Caenorhabditis* Genetics Centre for strains, funded by the NIH Office of Research Infrastructure Programs (P40OD010440).

## Additional information

### Funding

| Funder | Grant reference number | Author |
|---|---|---|
| H2020 European Research Council | ERC-StG-679990 | Sebastian Greiss |
| Medical Research Council | MC_U105181009 | Jason W Chin |
| Medical Research Council | MC_UP_A024_1008 | Jason W Chin |
| Wellcome Trust | Wellcome-Trust University of Edinburgh Institutional Strategic Support Fund ISS2 | Sebastian Greiss |
| Royal Society | | Sebastian Greiss |
| Muir Maxwell Epilepsy Centre | | Sebastian Greiss |
| Louis Jeantet Foundation | | Jason W Chin |
| University of Cambridge | Herchel Smith Fund | Inja Radman |
| University of Edinburgh | Edinburgh Global Award and Principal's Career Development PhD studentship | Zhiyan Xi |

The funders had no role in study design, data collection and interpretation, or the decision to submit the work for publication.

### Author contributions

Lloyd Davis, Data curation, Formal analysis, Investigation, Methodology, Visualization, Writing – original draft, Writing – review and editing; Inja Radman, Angeliki Goutou, Investigation, Writing – review and editing; Ailish Tynan, Investigation, Writing – review and editing, performed fluorescence microscopy; Kieran Baxter, Data curation, Investigation, Writing – review and editing, assisted with behavioural analysis; Zhiyan Xi, Investigation, Writing – review and editing, generated expression constructs and contributed to optimising ncAA incorporation; Jack M O'Shea, Investigation; Jason W Chin, Conceptualization, Funding acquisition, Supervision, Writing – original draft, Writing – review and editing; Sebastian Greiss, Conceptualization, Funding acquisition, Investigation, Methodology, Project administration, Supervision, Writing – original draft, Writing – review and editing

### Author ORCIDs

Jack M O'Shea ![ORCID] http://orcid.org/0000-0002-9694-7340
Sebastian Greiss ![ORCID] http://orcid.org/0000-0001-9130-0831

### Decision letter and Author response

Decision letter https://doi.org/10.7554/eLife.67075.sa1
Author response https://doi.org/10.7554/eLife.67075.sa2

## Additional files

### Supplementary files

• Supplementary file 1. Expression constructs and source plasmids used in this study.

• Transparent reporting form

### Data availability

All data are included in the manuscript and supporting files. Sequences used for generating transgenic constructs are listed in the supporting files.

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
