## [Decision Letter]

**Acceptance summary:**

This paper describes a new method for precise spatiotemporal control of gene expression in *C. elegans*. The authors express a photocaged Cre recombinase via incorporation of a photoactivatable amino acid through genetic code expansion, followed by uncaging in defined cells via optical targeting. They use this method to demonstrate distinct contributions of the left/right members of a bilateral neuron pair to mechanosensory behavior. With suitable validation in multiple tissue types, and confirmation that the endogenous physiology of the expressing cells remains unaffected as a consequence of the genetic code manipulation, this method could be broadly applicable.

**Decision letter after peer review:**

Thank you for submitting your article "Precise optical control of gene expression in *C. elegans* using genetic code expansion and Cre recombinase" for consideration by *eLife*. Your article has been reviewed by 2 peer reviewers, and the evaluation has been overseen by a Reviewing Editor and Piali Sengupta as the Senior Editor. The following individual involved in review of your submission has agreed to reveal their identity: Henrik Bringmann (Reviewer #2).

As you can see in the detailed reviews below, both reviewers are very appreciative of your findings, but also identified a number of shortcomings that will need to be addressed. Apart from some easily addressable editorial issues, there is a lack of quantification of the some of the results that we ask you to address. These suggestions are doable within the scope of a revision at *eLife* but the reviewers agree to give you some degree of freedom as to how you would like to incorporate these quantifications into your manuscript.

*Reviewer #1 (Recommendations for the authors):*

Even though some cells are genetically indistinguishable, many of them are visually distinguishable; thus, optical labeling and/or modulation of these cells are effective methods to distinguish individual cells. The authors have used two light-dependent techniques for the precise spatio-temporal regulation of single-cell activity: the light-dependent activation of the Cre transcription factor in a single neuron (spatial regulation) to express the light-dependent cation channel Chrimson and the light-dependent activation of Chrimson in the neurons of behaving worms at specific times (temporal regulation). Moreover, to achieve the clear activation of Cre, which had been difficult to achieve in the past, they have used the genetic code expansion technique in transgenic strains of an animal species for the first time (if I have understood correctly).

In my opinion, this method would be ideal for the precise spatio-temporal regulation of protein activity (and as a result, cell activity) if it works as designed. The biggest potential barrier would be the equal and effective incorporation of non-conventional amino acids (ncAAs) into all the cells in the animal as well as into the targeted protein. The authors have made a number of efforts and have reasonably evaluated the incorporation of ncAAs into the protein, although the evaluation of the incorporation of ncAAs into different types of cells is, unfortunately, far from satisfactory.

In addition, the data presentation is, in many cases, not quantitative; for example, sample numbers are not properly described, and only one picture per condition is provided for gene expression analysis. These need to be addresses to ensure reproducibility and general application of the results.

1) For the system to work, PCKRS, mt tRNA (C15), and Cre (TAG amber) need to be properly expressed in the cells. In addition, ncAAs should also be incorporated into the cells. I would like to see if this was actually observed in most, if not all, of the cells. Examples have been shown in Figure 2d and 2e, but these photos depict only a single worm per condition. Please analyze many more worms (I would recommend {greater than or equal to}10 worms per condition in total, including those from multiple transgenic lines grown on different days) and show the results in the form of a bar graph or a table. Additionally, it will be greatly appreciated if the authors are able to show whether the system works equally in different tissues, such as neurons, intestines, muscles, and hypodermis.

2) Related to above: Does the expression of these genes, in addition to ncAA incorporation, affect worm behavior? I think that addressing this question is important because truncated Cre is likely to be expressed in an amount {greater than or equal to}20 times higher than that of full-length PC-Cre (lines 250-256, Figure 2f). I would appreciate it if the authors could compare some basic aspects of worm locomotion, such as velocity and turning frequency, between the wild-type and the transgenic strains.

3) In this study, the clear-cut regulation of the activity of the gene product is critical. Nevertheless, the results have not been provided in a quantitative form. For example, the western blot shown in Figure 2c is, again, just one representative result. It should be repeated several times and the band intensities should be quantified and expressed as mean {plus minus} SEM, as shown in Figure 2f and 2g. (For Figure 2f, please describe the number of replicates).

4) Similarly, some data are shown only in the video. As mentioned previously, they should be reported using graphs showing the results for a certain number of samples (e.g., lines 277-279, Sup. video 1, 2).

5) lines 383-385 "We saw strong, easily visible expression": It was not clear which data support this statement. Further, describe how many worms were analyzed and how many worms were fluorescence-positive.

6) "more than 50 fold" and "close to 100%" (line 442): Again, these statements were not supported by evidence. At least, the link with specific data was not clear at all. These non-quantitative statements should be reworded to be more quantitative.

7) As mentioned above, in the figure legends, please describe how many worms were studied and how many were fluorescence-positive, instead of the number of replicates. The description provided in lines 637-640 is insufficient.

8) Sup. Table 2 (strains): Please specify the figure panels in which each strain was used.

*Reviewer #2 (Recommendations for the authors):*

The Greiss and Chin labs developed a system for optically activating Cre recombinase specifically in neurons that are exposed to UV-laser light, providing a tool to control gene expression in individual neurons for which there is currently no specific genetic targeting possible. They use a clever combination of incorporation of a photoactivatable artificial amino acid through genetic code expansion with photoactivation of Cre recombinase through precise optical targeting. They demonstrate the feasibility of their approach by targeting command interneurons and individually each neuron of a left-right pair of mechanosensory neurons for optogenetic activation. They demonstrate that the two mechanosensory PLM neurons have synergistic roles in the response to mechanical sensation but that each neuron plays a distinct role in the processing of the mechanical stimulus.

The method apparently is more complicated to carry out than just using a transgenic strain directly off the plate, with the expression of several components, feeding an unnatural amino acid, and the need to optically target each individual cell manually to achieve expression. Despite this extra effort that is required to perform a genetic targeting experiment, however, this method will be useful when each of a pair of two individual neurons needs to be investigated. Future developments of the method might help simplify the procedure. The effects of the complex manipulations on the neurons and their physiology should be investigated by control experiments.

Overall, the method of targeting individual neurons in a model organism is of substantial use for the *C. elegans* community. Likely there will be much to be learned from targeting individual neurons. Also, the method should be transferable to other genetically accessible and transparent systems such as zebrafish, and could thus become of broader use.

In summary, this is a very elaborate and smart approach to address an important technical problem in neuroscience and I support publication. I have the following comments:

1. To help other labs getting started with this method there should be a supplementary detailed step-by-step protocol for how to carry out the procedure.

2. It is not quite clear why the PLM neurons show different responses to stimulation. Is this caused by gene expression or connectivity? This could be discussed more deeply. It might help to draw a circuit diagram for the PLM and motor circuit that shows the differences of the two neurons.

3. The authors should discuss the effects that potential read-through of endogenous termination signals might have. How would this affect the physiology of the cell, especially during development? Would it theoretically be necessary and if yes feasible that some or all of the amber codons are replaced by TAA codons in future studies? This could be discussed.

4. The expression of the orthogonal translation system and the potential read-through of amber stop codons in endogenous genes poses a concern as to the effects on the physiology of the neurons. To mitigate such concerns regarding a new method it would be important to test whether after the complex manipulations the neurons have retained most of their physiological characteristics. Both the glr-1 command interneurons and mechanosensitive neurons are well characterized in terms of behavior and calcium activity. Thus, control experiments should be performed that test for the functionality of these cells. For example, do the worms still respond normally to anterior and posterior touch based on behavioral and calcium activity readout?

5. At present, the text is quite complicated to read, in particular for readers that want to only use the method and might not be interested in how the many technical challenges were overcome. Thus, some of the description of the optimization in the results part could go into the methods section and in the main text only a short summary of what was optimized could be described. This would help make this piece more accessible to a broad readership.

6. The authors establish the photoactivatable Cre system for glr-1-expressing neurons. They do not test, however, whether this strain allows cell-type specific activation. The glr-1 promoter expresses in both forward (such as PVC) and reverse command interneurons (Such as AVA) (Brockie et al. 2001). But a specific promoter for PVC that expresses only in this neuron is still not reported in the literature. It should be easy for the authors to activate only PVC and demonstrate a forward escape response when illuminating the entire animal. This would provide a proof of principle that their method can allow expression in a type of neurons that has been previously refractory to specific genetic control and would underscore the importance of the method.

---

## [Author Response]

Reviewer #1 (Recommendations for the authors):1) For the system to work, PCKRS, mt tRNA (C15), and Cre (TAG amber) need to be properly expressed in the cells. In addition, ncAAs should also be incorporated into the cells. I would like to see if this was actually observed in most, if not all, of the cells. Examples have been shown in Figure 2d and 2e, but these photos depict only a single worm per condition. Please analyze many more worms (I would recommend {greater than or equal to}10 worms per condition in total, including those from multiple transgenic lines grown on different days) and show the results in the form of a bar graph or a table. Additionally, it will be greatly appreciated if the authors are able to show whether the system works equally in different tissues, such as neurons, intestines, muscles, and hypodermis.

In Figure 2D we show the subcellular localisation of a PCKRS::GFP fusion (not incorporation). The purpose of this figure is to show the effect of the 4 different nuclear export sequences (NES) on subcellular localisation. We have now included quantification of nuclear localisation by measuring the nuclear to cytoplasmic ratio for the wild type and the 4 NES constructs. Measurements were taken from 2 independent lines for each construct (Figure 2 —figure supplement 1B).

The images in Figure 2E are representative images, showing > 5 randomly selected animals for each condition, we now state in the legend that the animals were randomly selected. Furthermore, all western blots were done using lysates from randomly selected animals. The quantitative western blots in Figure 2F were done using the lines shown in 2E, as well as a further independent line for each genotype. The western blot shown in Figure 2G was also done using the same lines shown in 2E. The incorporation experiments in 2F were performed in triplicate (performed on different days) and each sample was blotted and measured twice. This information is now included in the figure legend. We have added as a supplement to Figure 2 an enlarged version of the "PKIa-NES/tRNA(C15)" and "Smad4-NES/tRNA(C15)" panels in figure 2E to make it more apparent that we show multiple animals in the image.

We and others have previously published papers describing genetic code expansion in *C. elegans* (Greiss and Chin, JACS 2011; Parrish et al., ACS Chemical Biology 2012), showing that ncAA incorporation works in different tissues. We now refer to this in the discussion and reference the previous work. In the enlarged panels from Figure 2E, which we have added as a supplement (Figure 2 —figure supplement 1C), it is clearly visible that incorporation occurs globally, in multiple cell types. Furthermore, genetic code expansion has, since we first demonstrated it in C. elegans, been established for several multicellular organisms including fruit fly, zebrafish, mouse, Arabidopsis, where incorporation of ncAA has also been demonstrated in different tissues for these organisms. Our manuscript presents modifications to the pyrrolysyl-tRNA-synthetase / tRNA pair, which is central to the genetic code expansion method, and we show that we thereby improve efficiency of the pair within the context of the C. elegans translational machinery. We expect this basic machinery to be the same in all C. elegans cells, albeit factors such as ncAA availability and expression levels of the genetic components may vary between cell types. We quantify the improvements to the system directly, using a fluorescent GFP::mCherry reporter that is expressed using a ubiquitous promoter, and by assessing its efficacy as a biological tool in different neuronal cell types.

2) Related to above: Does the expression of these genes, in addition to ncAA incorporation, affect worm behavior? I think that addressing this question is important because truncated Cre is likely to be expressed in an amount {greater than or equal to}20 times higher than that of full-length PC-Cre (lines 250-256, Figure 2f). I would appreciate it if the authors could compare some basic aspects of worm locomotion, such as velocity and turning frequency, between the wild-type and the transgenic strains.

We have included a soft touch assay performed on the lines used for the behavioural assays shown in Figure 5, which express the genetic code expansion machinery and photocaged Cre in the touch receptor neurons. The touch assays were performed in the presence and in the absence of PCK, showing that neuronal function is not affected by the expression of truncated Cre, nor by any effect of PCK incorporation. We also performed touch assays 24h after uncaging of animals grown on PCK. We include these data in Figure 4 —figure supplement 1. With our assays we show that it is clearly possible to express the genetic code expansion machinery and photocaged Cre recombinase at sufficient levels for the system to work efficiently without inhibiting cellular function.

3) In this study, the clear-cut regulation of the activity of the gene product is critical. Nevertheless, the results have not been provided in a quantitative form. For example, the western blot shown in Figure 2c is, again, just one representative result. It should be repeated several times and the band intensities should be quantified and expressed as mean {plus minus} SEM, as shown in Figure 2f and 2g. (For Figure 2f, please describe the number of replicates).

We have now included quantification of incorporation efficiency for the genotypes in Figure 2C, which we present in Figure 2 —figure supplement 1A.

Our rationale for originally not quantifying incorporation of all genotypes shown in 2C is as follows:

The first 5 genotypes in Figure 2C show the first step of our optimisation, namely finding the optimal NES sequence. Since two of the four nuclear export sequences tested appeared to reduce efficiency based on the western blot, we focused on the two best candidates for the quantification and further combined them with the optimised tRNAs.

We now also include the numbers of replicates for Figure 2C (each line was quantified in two independent experiments on different days) and Figure 2F (each line was quantified in three independent experiments on different days, and each lysate was measured twice on different blots). For Figure 2F we assayed two independent lines for each condition shown.

4) Similarly, some data are shown only in the video. As mentioned previously, they should be reported using graphs showing the results for a certain number of samples (e.g., lines 277-279, Sup. video 1, 2).

We have included a quantification of the behavioural response shown in Videos 1 and 2 in Figure 3G and in Figure 3 —figure supplement 1C. In addition to the line shown in the videos we have included lines expressing the non-optimised incorporation system and non-optimised PC-Cre. We assayed two independent lines for each condition in two independent experiments each. We have added a reference to the figures when we reference the videos.

5) lines 383-385 "We saw strong, easily visible expression": It was not clear which data support this statement. Further, describe how many worms were analyzed and how many worms were fluorescence-positive.

We have rephrased the sentence to make clear that the fluorescence from Chrimson::mKate2 appeared after 12-24h and that it appeared in the targeted neurons, as shown in Figure 4 C,D,E. It now reads:

" After 12-24h, we observed Chrimson::mKate2 expression that was only present in laser-targeted cells."

We have removed the sentence stating that the fluorescence was also visible under a Leica dissecting fluorescence microscope. We have moved this sentence to the methods section. where we describe the preparation of animals for behavioural experiments (feeding on PCK -> uncaging -> verification of Chrimson::mKate2 expression using a Leica fluorescence dissecting microscope).

6) "more than 50 fold" and "close to 100%" (line 442): Again, these statements were not supported by evidence. At least, the link with specific data was not clear at all. These non-quantitative statements should be reworded to be more quantitative.

We have removed these numbers from the discussion. They can be found in the Results section closer to the relevant figures (2F and 3D) (page 6 and page 8).

7) As mentioned above, in the figure legends, please describe how many worms were studied and how many were fluorescence-positive, instead of the number of replicates. The description provided in lines 637-640 is insufficient.

We have now included descriptions in the figure legends.

8) Sup. Table 2 (strains): Please specify the figure panels in which each strain was used.

We have added a key resources table with the information. It can be found at the beginning of the methods section.

Reviewer #2 (Recommendations for the authors):In summary, this is a very elaborate and smart approach to address an important technical problem in neuroscience and I support publication. I have the following comments:1. To help other labs getting started with this method there should be a supplementary detailed step-by-step protocol for how to carry out the procedure.

We have previously published a detailed protocol in a volume of Methods in Molecular Biology: Davis, L. and Greiss, S. (2018) ‘Genetic encoding of unnatural amino acids in *C. elegans*’, Methods in Molecular Biology, 1728, pp. 389–408. doi: 10.1007/978-1-4939-7574-7_24. We have now included references to this book chapter in the sections of the methods where we describe generating transgenic strains, ncAA incorporation, and uncaging.

2. It is not quite clear why the PLM neurons show different responses to stimulation. Is this caused by gene expression or connectivity? This could be discussed more deeply. It might help to draw a circuit diagram for the PLM and motor circuit that shows the differences of the two neurons.

We have included a circuit diagram in Figure 5C. While the diagram does not include all synaptic connections between the neurons depicted, we focused on the main connections and the differences between PLML and PLMR. We believe that including all connections would make the diagram very difficult to interpret. We include a paragraph in the discussion (page 14) about the different responses we observe for PLML and PMR, which we indeed believe to be due to differences in connectivity.

To exclude the possibility that the responses may be due to differences in expression levels, we have added Figure 5 —figure supplement 3 showing quantification of expression in PLML and PLMR, determined by quantifying the mKate2 fluorescence of the Chrimson::mKate2 fusion protein. We see no differential expression of the optogenetic channel.

3. The authors should discuss the effects that potential read-through of endogenous termination signals might have. How would this affect the physiology of the cell, especially during development? Would it theoretically be necessary and if yes feasible that some or all of the amber codons are replaced by TAA codons in future studies? This could be discussed.

We have included the possibility of read-through of endogenous stop codons in the discussion (2nd paragraph in the discussion, page 13) and we have added a reference to a paper by the Fire lab, describing a natural mechanism employed by *C. elegans* to mitigate against stop codon read-through: Arribere, J. A. et al., (2016) ‘Translation readthrough mitigation’, Nature. Nature Publishing Group, pp. 1–17. Available at:

http://dx.doi.org/10.1038/nature18308.

We would expect that this mechanism also mitigates against any read-through effects caused by the orthogonal system.

4. The expression of the orthogonal translation system and the potential read-through of amber stop codons in endogenous genes poses a concern as to the effects on the physiology of the neurons. To mitigate such concerns regarding a new method it would be important to test whether after the complex manipulations the neurons have retained most of their physiological characteristics. Both the glr-1 command interneurons and mechanosensitive neurons are well characterized in terms of behavior and calcium activity. Thus, control experiments should be performed that test for the functionality of these cells. For example, do the worms still respond normally to anterior and posterior touch based on behavioral and calcium activity readout?

We opted to address this point by quantifying behavioral readouts: we have performed soft touch assays to test for the functionality of the cells and we find that the touch receptor neurons still function normally, both in the absence and in the presence of PCK. We have also included animals grown on PCK and assayed 24h after uncaging, i.e. the exact treatment and timepoint we use to conduct the behavioural assays. The data are included in Figure 4 —figure supplement 1C.

In both neuronal classes, the TRN and glr-1, we can elicit the behaviour expected from the literature when optogenetically stimulating them. Calcium imaging would not be feasible in the strains we describe as we have the optPC-Cre in an operon with GFP and we express the optogenetic channel as an mKate2 fusion.

5. At present, the text is quite complicated to read, in particular for readers that want to only use the method and might not be interested in how the many technical challenges were overcome. Thus, some of the description of the optimization in the results part could go into the methods section and in the main text only a short summary of what was optimized could be described. This would help make this piece more accessible to a broad readership.

We appreciate that the manuscript is quite technical at times, and in the resubmitted version we have tried to edit it for clarity.

The readership of the paper will consist of researchers looking to use photocaged Cre as a tool in *C. elegans*, but also researchers working in the genetic code expansion field or looking to use genetic code expansion in other applications. While we agree that for *C. elegans* groups wishing to use photocaged Cre, the first part of the paper may be of less interest, for those working in the genetic code expansion field, the first part may be more interesting than the second part. We would therefore prefer to keep a more detailed description of our optimisation approach in the main paper as it constitutes a major advance in the field.

6. The authors establish the photoactivatable Cre system for glr-1-expressing neurons. They do not test, however, whether this strain allows cell-type specific activation. The glr-1 promoter expresses in both forward (such as PVC) and reverse command interneurons (Such as AVA) (Brockie et al. 2001). But a specific promoter for PVC that expresses only in this neuron is still not reported in the literature. It should be easy for the authors to activate only PVC and demonstrate a forward escape response when illuminating the entire animal. This would provide a proof of principle that their method can allow expression in a type of neurons that has been previously refractory to specific genetic control and would underscore the importance of the method.

We chose the PLM neurons as our proof-of-principle since they also cannot be targeted using a specific promoter. Using the PLMs, we have clearly demonstrated that we are able to target a neuronal pair, as well as the individual members of the pair.

Furthermore, we believe that due to the asymmetric downstream connectivity of PLML and PLMR, they represent an attractive first target for single cell optogenetics. We fully agree however, that the PVCs will indeed be interesting targets in the future.